# MultiHal: MultiLingual Dataset for Knowledge-Graph Grounded Evaluation of LLM Hallucinations

**Ernests Lavrinovics** [1]   **Russa Biswas** [1]   **Katja Hose** [2]   **Johannes Bjerva** [1]

## Abstract

Large Language Models (LLMs) have inherent limitations of faithfulness and factuality, commonly referred to as hallucinations. Several benchmarks have been developed that provide a test bed for factuality evaluation within the context of English-centric datasets, while relying on supplementary informative context like web links or text passages but ignoring the available structured factual resources. To this end, Knowledge Graphs (KGs) have been identified as a useful aid for hallucination mitigation, as they provide a structured way to represent the facts about entities and their relations with minimal linguistic overhead. We bridge the lack of KG paths and multilinguality for factual language modeling within the existing hallucination evaluation benchmarks and propose a KG-based multilingual, multihop benchmark called **MultiHal** framed for generative text evaluation. As part of our data collection pipeline, we mined 140k KG-paths from open-domain KGs, from which we pruned noisy KG-paths, curating a high-quality subset of 25.9k. Our baseline evaluation shows an absolute scale improvement by approximately 0.12 to 0.36 points for the semantic similarity score, 0.16 to 0.36 for NLI entailment and 0.29 to 0.42 for hallucination detection in KG-RAG over vanilla QA across multiple languages and multiple models, demonstrating the potential of KG integration. We anticipate MultiHal will foster future research towards several graph-based hallucination mitigation and fact-checking tasks.

**Code:** github.com/ernlavr/multihal

**Data:** huggingface.co/datasets/ernlavr/multihal

[1]Department of Computer Science, Aalborg University, Copenhagen, Denmark [2]Institute of Logic and Computation, TU Wien, Vienna, Austria. Correspondence to: Ernests Lavrinovics <elav@cs.aau.dk>.

*Proceedings of the $43^{rd}$ International Conference on Machine Learning*, Seoul, South Korea. PMLR 306, 2026. Copyright 2026 by the author(s).

## 1. Introduction

Factual inconsistencies in LLM outputs, commonly referred to as hallucinations, are often a bottleneck for production-grade deployment of LLM systems (Huang et al., 2025a). Although hallucinations may be beneficial for tasks involving creativity (Jiang et al., 2024) or even drug discovery (Yuan et al., 2025), they become a liability for other tasks that require factually consistent outputs, for example, information retrieval, summarization and question answering (Lavrinovics et al., 2025). Additionally, Huang et al. (2025a); Augenstein et al. (2024) suggests that hallucinations impair the trust and usefulness of AI systems, and even pose certain societal risks by enabling the generation of convincing misinformation (Augenstein et al., 2024; Puccetti et al., 2024). Hallucinations can stem from multiple shortcomings in model training, such as reinforcement learning from human feedback (RLHF) (Bai et al., 2022), in cases when human preferences are towards non-factual answers (Zhang et al., 2025), instruction tuning where given instructions exceed a model's knowledge boundary (Zhang et al., 2025; Huang et al., 2025b), or due to lack of up-to-date knowledge. Furthermore, hallucinations occur with varied levels of frequency and intensity depending on the generated language (Chataigner et al., 2025; Qi et al., 2023). A general trend is observed that, in terms of factual consistency, English outputs are the most stable and overall factual quality decreases with lower resourced languages. This varied degree of factuality across languages only further impairs the usability and inclusiveness of LLMs in different applications.

To this end, Retrieval Augmented Generation (RAG) (Niu et al., 2024; Zhao et al., 2026) is the most widely adopted method for improving factuality, which supplements the input query to an LLM with relevant text passages to improve the factuality of LLM outputs. The main advantage of RAG is that it does not require retraining the generator LLM, a process that is time-consuming and resource-intensive. However, RAG is still limited by the LLM context window size (Zhao et al., 2026), its sensitivity to input prompt formatting (Mizrahi et al., 2024; Maia Polo et al., 2024), and *Needle in a Haystack* problem (Gao et al., 2026), where important details can be lost in a large pool of text.

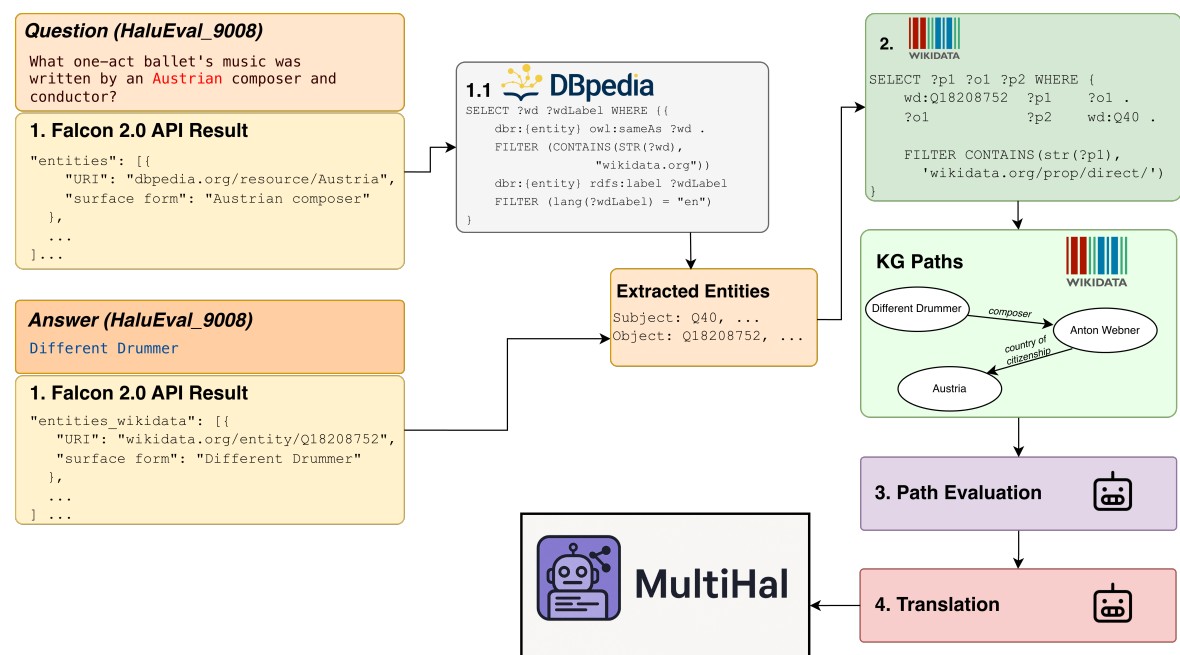

*Figure 1.* Overview of MultiHal pipeline with example data point *HaluEval_9008*. The pipeline's sequential steps are enumerated, Step 1.1 is an auxiliary step that maps DBpedia entities to Wikidata.

KG-RAG (Sanmartín, 2024; Peng et al., 2025) provides several advantages over document-based RAG and has also been suggested as a promising methodology for limiting LLM hallucinations (Pan et al., 2024; 2023), primarily leveraging the structural and factual qualities of the KGs through sets of KG-paths that describe entities and their relationships with minimal linguistic overheads. Furthermore, KG integration within language modeling can alleviate the need for full re-training when utilized during inference (Sun et al., 2023; Luo et al., 2024) or post-generation (Guan et al., 2024). This is valuable for use-cases with rapidly developing knowledge or limited computational resources (Lavrinovics et al., 2025). The structured, factually rich and linguistically minimal qualities of the KGs can potentially decrease the risks of the *Needle-in-a-Haystack* problem and limitations of the context window size. Conditioning LLMs on KGs can also enable optimal output scrutiny and explainability by allowing the outputs to be traced back to explicit sources, making cross-checking less time-consuming than document-based RAG. Furthermore, KGs accompany each entity with rich metadata, but their optimal use in factual language modeling is still an open question.

Although KG-RAG is rapidly gaining attention to improve the factuality in LLM, existing QA benchmark data sets (Zhao et al., 2023; Lin et al., 2022; Li et al., 2023; Wei et al., 2024; Rahman et al., 2025; Mickus et al., 2024; Ravi et al., 2024; Bang et al., 2025) on LLM hallucinations rely primarily on textual data for contextual information and provide no multilingual support. While the questions

in these benchmark datasets are compiled from different sources, the answers for FELM (Zhao et al., 2023) HaluEval (Li et al., 2023) Shroom2024 (Mickus et al., 2024) are LLM-generated and evaluated using LLM-as-a-judge or human annotation. For some datasets (Lin et al., 2022; Zhao et al., 2023; Wei et al., 2024), the answers are supported with external contextual information from textual resources such as webpages. Therefore, in this paper, we bridge these critical gaps by presenting a novel multilingual hallucination benchmark *MultiHal*, grounded on factual information from Wikidata (Vrandečić & Krötzsch, 2014) KG. *MultiHal* is based on a total of 7 common benchmarks that lack structured factual and multilingual coverage, namely **Felm** (Zhao et al., 2023), **TruthfulQA**(Lin et al., 2022) (TQA), **HaluEval**(Li et al., 2023), **HaluBench** (Ravi et al., 2024), **SimpleQA** (Wei et al., 2024), **DefAn** (Rahman et al., 2025), **Shroom2024** (Mickus et al., 2024). We propose a data collection framework as illustrated in Figure 1, to aggregate over 31k unique questions from aforementioned datasets, enriching them by mining 140k KG paths and ensuring factual consistency by filtering using LLM-as-a-judge. To enable multilingual hallucination evaluation, our compiled dataset comprising questions, ground-truth answers and KG paths, is translated to Spanish, French, Italian, Portuguese and German. Therefore, our main contributions are as follows:

1. We present a multilingual, multi-hop factual language modeling benchmark grounded with information from KGs which we call **MultiHal**. The code and data are made publicly available.

2. We propose a novel unified scalable framework that systematically integrates entity linking methods, mapping question-answer pairs to a KG, to curate factual information from KGs.

3. To support a robust multilingual evaluation, we provide high-quality translations of the question-answer pairs and their corresponding KG paths in 5 different languages.

4. We evaluate the quality of KG path filtering based on LLM-as-a-judge by analyzing their correlation with the semantic scores between predicted and gold answers for each question.

5. Baseline experiments reporting on the semantic similarity of LLM models in vanilla QA and KG-RAG based settings, demonstrating the effectiveness of incorporating KG paths.

**Conflict of Interest Disclosure:** We declare **no** financial conflict of interest arising as part of this work. Funding sources are further acknowledged in the Acknowledgments section.

## 2. MultiHal

MultiHal builds upon a set of 7 previously established benchmarks by enriching them with factual information in the form of relevant paths from Wikidata. The choice of these benchmarks is motivated by their relevancy to factuality evaluation, yet they lack support for factual grounding of the answers, leveraging KG and LLM integration models, and multilingual evaluation. We summarize the basic dataset statistics in Table 1, for MultiHal a dataset schema description see Appendix A. These foundational benchmarks are all filtered for generative question-answering based on general/trivia domains. Furthermore, benchmarks such as Shroom2024 (Mickus et al., 2024), FELM (Zhao et al., 2023), HaluEval (Li et al., 2023), HaluBench (Ravi et al., 2024) are primarily oriented towards evaluating hallucination detection models consisting of both *hallucinated* and *non-hallucinated* data, therefore, the data is repurposed by filtering for rows labelled as *non-hallucinated*. We consider that each unique *question-path* pair as a data point. The count difference between data points and unique questions is due to multiple candidate paths per question. The overview of each of the processing stages in our dataset collection pipeline is illustrated in Figure 1. The following sections scope into the methodological details of each of the processing stages of the proposed dataset collection framework. Additionally we report on our computing processing times and CO2 emissions in Appendix J. Our original contributions are released under CC-BY-4.0 license terms.

### 2.1. Dataset Preprocessing

Considering that *MultiHal* builds upon established benchmarks, question deduplication is performed to avoid data leakage across the foundations. Deduplication is based on computing sentence embeddings using SentenceTransformers[1] (Reimers & Gurevych, 2019) and computing all possible pair-wise cosine similarity between the questions. The ground-truth answers and any present supplementary context of the pair of questions with a sentence similarity threshold above **0.99** are merged. Deduplication was exclusively skipped for *DefAn-QSRanking* subset due to a large amount of questions consisting of nearby years for corresponding university rankings, which led to a very high number of false positives among the data points.

Additionally, we discard data points where the ground-truth answers are phrases such as *"I have no comment"*, which indicate refusal to answer, and we define them as *refusal types*. We compile a list of refusals consisting of a list of text patterns as described in B. Any rows with output columns that exactly match, case-insensitively, one of these refusal phrases are filtered out.

### 2.2. KG Path Mining

The overall idea is to mine relevant paths from Wikidata (Vrandečić & Krötzsch, 2014). The core semantic entities are extracted from a given question $Q$ and its ground-truth answer $A$, and afterward matched to Wikidata entities. The extracted entities in $Q$ and $A$ are used for querying Wikidata for existing paths.

**Entity Matching from Text to Knowledge Graphs.** The core entity extraction and matching from raw text is based on Falcon 2.0 (Sakor et al., 2020). Falcon 2.0 is an open-sourced framework which is also made available via an API[2] that we call to retrieve subjects from question $Q$ and objects from answer $A$. Given a text passage, Falcon 2.0 outputs a ranked list of entities as candidates in Wikidata, we use the Top-3 candidates. For increased redundancy, we use Falcon 2.0 to additionally return DBpedia entities, which we then map back to Wikidata using the query in Listing 1 in Appendix C. Additionally, foundational benchmarks such as *FELM* (Zhao et al., 2023), *SimpleQA* (Wei et al., 2024), *TruthfulQA* (Lin et al., 2022) contain supplementary context in the form of Wikipedia links which we map to Wikidata(Vrandečić & Krötzsch, 2014) entities using Wikipedia public API [3].

The *Wikipedia-to-Wikidata* retrieval is done by taking the

---

[1]https://huggingface.co/sentence-transformers/all-MiniLM-L6-v2

[2]https://labs.tib.eu/falcon/falcon2/

[3]https://en.wikipedia.org/w/api.php?action=query &prop=pageprops&titles=$WIKIPEDIA_ID&format=json

*Table 1.* Compositional statistics of MultiHal for a single language. †HaluBench includes HaluEval, hence excluded to avoid data leakage. ‡ Paraphrasings of each question in DefAn are also discarded.

| Dataset | Subset | License | Data points (unique paths) | Unique questions | Domains | Question length (char) | Answer length (char) |
|---|---|---|---|---|---|---|---|
| HaluEval | QA | MIT | 11,398 | 3420 | 1 | 115.46 | 13.95 |
| HaluBench | Whole except HaluEval† | CC-by-nc-2.0 | 626 | 200 | 4 | 105.73 | 272.72 |
| Defan | Whole‡ | MIT | 9,969 | 1975 | 5 | 93.48 | 13.31 |
| SimpleQA | Whole | MIT | 3300 | 1246 | 10 | 86.97 | 11.14 |
| TruthfulQA | Generative | Apache 2.0 | 193 | 77 | 26 | 76.15 | 37.11 |
| Shroom2024 | Definition Modeling | CC-BY | 346 | 160 | 1 | 170.86 | 73.37 |
| Felm | World knowledge | CC-BY-NC-SA-4.0 | 73 | 17 | 1 | 95.25 | 75.26 |
| MultiHal (total) | - | CC-BY-4.0 | 25,905 | 7095 | 48 | 106.27 | 70.98 |

page title embedded in the given Wikipedia link and replacing it with the *$WIKIPEDIA_ID* placeholder. The *Top-3 candidates*, *DBpedia-to-Wikidata* and *Wikipedia-to-Wikidata* processing steps are all done for redundancy purposes to increase the chances of retrieving high quality KG paths.

**Knowledge Graph Querying.** We query Wikidata in order to find existing paths between the extracted *subject-object* entities up to 2 hops. As additional pre-processing steps before querying, we remove circular *subject-object*, as well as create an inverted set of *subject-object* pairs to accommodate for the directionality of the Wikidata graph. Depending on the foundational benchmark, we create custom queries for the different answer types we encounter when merging all our foundational benchmarks. The answer type is denoted by *answer_type* column in MultiHal, see the schema in Appendix A. In Appendix C, see Listing 2 for Wikidata entity query, Listing 3 for date-literal query and Listing 4 for numerical-literal query. The answer types, such as numericals and dates, are queried with value limitations for numerical and time-based properties in the final hop, as shown in Listings 4 and 3 to improve query speed. See Appendix D Listing 6 for a set of time-based properties and Listing 7 for numerical properties. For querying, we use the public Wikidata endpoint[4], our path cut-off date is April 2025.

For decoding the Wikidata entity labels, we run a separate pass using the query in Appendix C Listing 5. When querying for labels, we discard any statements, entities or objects that cannot be directly mapped to natural language text labels.

### 2.3. KG Path Quality Evaluation: LLM as Judge

As a method for filtering out noisy KG paths and identifying high-quality paths, we employ a two-step LLM-as-a-judge methodology (Li et al., 2025; Yu et al., 2024): firstly, for questions with more than 10 candidate paths the top-10 paths selection is done to limit the total count; secondly, scoring each path individually to identify low and high quality paths. For both selection and scoring we use **GPT-4o Mini** similarly to Laskar et al. (2025); Arif et al. (2025).

We further motivate the choice of GPT-4o-Mini in Section 4.1. For inference we use OpenRouter API[5] with sampling temperature **0.1**. The goal of the *selection* is to decrease the overall number of KG paths, resulting in a decrease from **140k** to **25.9k** paths.

**Selection Step.** We construct a prompt for selecting the 10 most relevant paths with respect to the question-answer, and available optional answer pairs. Selection is intended to be done without any particular ordering, see Listing 8 in Appendix E. The set of paths for each question is processed in two passes, and in both scenarios, the order of the paths is shuffled to avoid any ordering biases (Li et al., 2025). From the two passes, we consider only the overlapping paths from the two candidate sets as the final collection of paths. During the selection phase, LLM-generated outputs are validated by checking for exact matches against the set of paths corresponding to the question. Any generated paths that do not have an exact match are discarded to mitigate the risk of syntactic errors or hallucinations introduced by the LLM-as-a-judge as a method of quality control. This process is repeated up to three times or until a total of 10 valid paths are obtained. If, after three attempts, fewer than 10 valid paths are selected, the remaining slots are filled by randomly sampling from the original KG path pool for the corresponding question. The selection step is bypassed for questions that have 10 or fewer candidate paths.

**Scoring Step.** Once a set of candidate paths is established, we construct another prompt for rating their relevance with respect to the given question and answer, and we process each path individually, see Appendix E Listing 9 for the instructions. Scoring is done by determining the *quality score* on a scale of 1-5, where 1 indicates a path which is completely unrelated to the question and answer, and 5 indicates an explicit answer to the question. From our final benchmark we filter out all paths rated 1-3, which we deem as *low-quality* and leave only paths rated 4-5 as *high-quality* ones.

---

[4]https://query.wikidata.org/bigdata/namespace/wdq/sparql

[5]https://openrouter.ai/openai/gpt-4o-mini

## 2.4. Multilinguality

For enabling multilinguality for MultiHal, we employ the Nllb-200 3.3bn (Costa-Jussà et al., 2022) model and focus on its five well-performing European languages, namely *German*, *Italian*, *French*, *Portuguese* and *Spanish*. Our generation hyperparameters are specified in Table 2.

*Table 2.* Overview of Nllb200-3.3bn inference hyperparameters

| | |
|---|---|
| *Batch size* | 8 |
| *Decoding* | Beam search |
| *Beam size* | 5 |
| *Length penalty* | 1.1 |
| *Early stopping* | True |
| *No repeat ngram* | 2 |
| *Max sequence* | 1024 |

Empirically, we found these hyperparameters to work the most optimal for our use case. We also noted that by separating the labels with *semicolons* yielded more accurate translations than having KG path labels purely whitespace separated, we attribute this to the improper grammatical structures that occur when label entities are not separated. We observe that Nllb-200's output translations are generally of high quality, yet Nllb-200's model does not always correctly output semicolon separation between the entities and predicates with respect to the English source. For more details for human audits, see Appendix P.

## 3. Experimental Setup

The baseline experiments are set up using a prompt-based knowledge injection method. The prompt $\mathcal{P}$ is formatted as $\mathcal{P} = (\mathcal{K}, \mathcal{Q})$, where $\mathcal{K}$ is knowledge in the form of a KG path and $\mathcal{Q}$ is the question of the data point, see Appendix F Listing 10 for KG-RAG and Listing 11 for vanilla QA for used prompts. We conduct experiments with and without knowledge $\mathcal{K}$ (KG-RAG and vanilla QA respectively) to measure the effectiveness of the factual information contained in the KG paths. We measure the semantic similarity between ground-truths and model predictions using Multilingual-MiniLM-L12-v2[6] (Wang et al., 2024), the choice of the sentence embedding model is based on results in the MMTE benchmark (Enevoldsen et al., 2025), its multilingual capabilities, and comparatively small parameter count. Semantic similarity is computed by mean-pooling the last hidden states per token for each sentence, applying L2 normalization and computing the dot-product between the ground-truth and LLM prediction representations. For experimental conditions, we use Gemini 2.0 Flash, GPT-4o Mini, and Llama 3.3 70bn instruct models. Additionally we

---

[6]https://huggingface.co/sentence-transformers/paraphrase-multilingual-MiniLM-L12-v2

*Table 3.* Overview of preliminary baseline results for KG-RAG QA task with Gemini 2.0 Flash as answer generator. Results are based on dataset subsplits in Figure 2.

| Path Judge Model | SemScore | Correlation |
|---|---|---|
| GPT 4o-Mini | 0.513 | 0.485 |
| Gemini 2.0 Flash | 0.529 | 0.430 |

triangulate our semantic similarity results with NLI and hallucination detection. Inspired from Bang et al. (2025) we run hallucination detection on the **English** subsplit of MultiHal using HHEM-2.1 (Bao et al., 2024), and run NLI evaluation similar to Sansford et al. (2024); Zhang et al. (2024) based on a DeBERTa multilingual NLI model [7] namely due to model's language coverage and its performance. The NLI and HHEM-2.1 models expect test pairs as *hypothesis* and *premise*. We form the *premise* as a prompt containing a question and ground-truth $\mathcal{P} = \{\mathcal{Q}; \mathcal{G}\}$ and *hypothesis* is model response $\mathcal{A}$.

Additionally, we compute the Spearman correlation between the semantic similarity score and the *quality score* of each KG path, aiming to quantify the reliability of the quality score (see Section 2.3) determined by LLM-as-a-judge. Our assumption is that these quality scores should positively correlate with the computed semantic similarity score between the ground truth and the predicted answer in KG-RAG, i.e., when conditioned on the paths as supplementary information. For running the model prediction computations, we employ the OpenRouter API service[8] and perform the generation with sampling temperature set to **1**.

## 4. Results

### 4.1. Preliminary Baselines

Considering that our methodology primarily relies on an LLM-judge for filtering and rating the quality of KG paths, we conduct a preliminary baseline test for the **English** subset to observe the performance of LLM judges. We conduct the experiment using a proportionally sampled subset of MultiHal, see Figure 2 for the data distribution. The goal of this is to gain insights on the expected quality of the KG paths.

For the preliminary test, we compared KG paths selected and rated by *Gemini 2.0 Flash* and *GPT-4o mini*, which were afterwards tested in a KG-RAG setting with *Gemini 2.0 Flash* model and computing the correlation between path ratings and semantic similarity, our results are summarized in Table 3.

---

[7]https://huggingface.co/MoritzLaurer/mDeBERTa-v3-base-xnli-multilingual-nli-2mil7

[8]https://openrouter.ai/

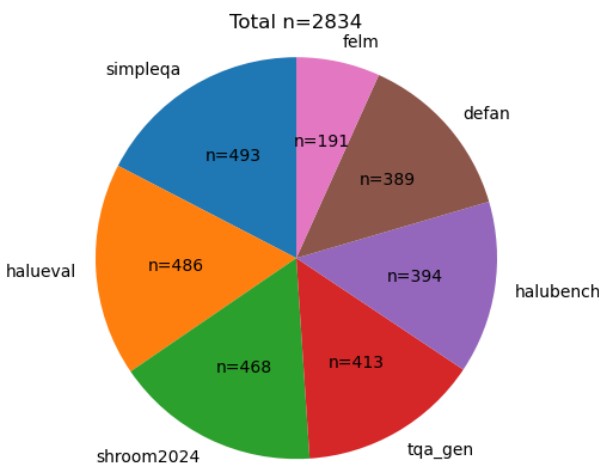

*Figure 2.* Breakdown of the data point count (n) distribution per foundational benchmark evaluated as part of the preliminary baseline experiment.

*Table 4.* GPT 4o-Mini overview of false positives, false negatives and IAA - all computed with respect to human judgement. Before computation all path scores are binarized as low (ratings 1, 2, 3) and high (ratings 4, 5) quality. Results are based on dataset subsplits in Figure 2.

| Metric | Value |
|---|---|
| False Positives | 11% |
| False Negatives | 2.78% |
| IAA Cohen-Kappa | 0.62 |

From the results in Table 3 we see that paths judged by **GPT 4o-Mini** have a higher correlation with the semantic similarity. Given that low quality paths (rated 1-3) impair the LLM output quality and high quality paths (4-5) improve it (see Appendix L), we chose to run the full baseline experiments with GPT 4o-Mini. For a more in-depth breakdown of GPT 4o-Mini performance per dataset and domain, please refer to Appendix G. Table 4 showcases false positives and false negatives for GPT 4o-Mini as well as interannotator agreement with respect to human judgment as ground truth, refer to Appendix Q for a numerical overview of false-positive distributions. IAA score between human and GPT 4o-Mini annotations, including a focus on misclassifications by GPT-4o which gives an indication of noise presence which is comparable to datasets and can be used for reproducibility.

### 4.2. Baseline Experiments

We report our baseline experiment results in Figure 3 for semantic similarity, Table 5 for aggregated NLI scores over languages and Table 6 for hallucination detection on the English subsplit, see Appendix N for a fine-grained overview of semantic similarity over domains and Appendix M for NLI results expanded over each language. The results show-

case a consistent improvement of KG-RAG setting over QA for all evaluation metrics, indicating that the mined KG paths are meaningful for a model to generate a higher quality output. The result distributions in Figure 3 have statistically significant differences across all languages and all models between the QA and KG-RAG conditions, see Appendix K for more details. For a full numerical overview of Figure 3, see Appendix H.Additionally, we run a follow-up experiment with an open-sourced LLM judge Qwen 2.5 72bn Instruct (Yang et al., 2025). The results are showcased in Appendix O. We release English-only Qwen 2.5 rated paths as a supplement to our main benchmark.

*Table 5.* Aggregated NLI results over all MultiHal languages. Ent is *entailment*, Neut is *neutral*, Contr is *contradicting*

| Model | Task | Ent | Neut | Contr |
|---|---|---|---|---|
| Gemini-2.0 | KG-RAG | 68% | 26% | 6% |
| Flash | QA | 52% | 24% | 24% |
| Llama-3.3- | KG-RAG | 71% | 21% | 8% |
| 70b instr | QA | 45% | 31% | 23% |
| GPT-4o Mini | KG-RAG | 76% | 16% | 8% |
| | QA | 40% | 32% | 28% |

*Table 6.* Result overview with hallucination detection using HHEM-2.1. *C* (consistent), *H* (hallucinated)

| Model | Task | C | H |
|---|---|---|---|
| Gemini-2.0 | KG-RAG | 87% | 13% |
| Flash | QA | 58% | 42% |
| Llama-3.3- | KG-RAG | 88% | 12% |
| 70b instr | QA | 55% | 45% |
| GPT-4o Mini | KG-RAG | 89% | 11% |
| | QA | 47% | 53% |

## 5. Ablation Study: Path Qualities

To further demonstrate the effectiveness of our mined paths, we conduct a study of baseline results for *Path Quality 4*. Results are presented in Table 7

The results showcase a decreased performance with respect to Figure 3 and Tables 5, 6 which is expected due to impaired path quality, yet the paths still consistently improve over vanilla QA and provide meaningful information for models to produce higher quality output. For further breakdown of path qualities we refer the reader to Appendix L.

## 6. Discussion

Overall, the results from Figure 3 and Tables 5, 6 depict a consistent improvement in a KG-RAG for all tested LLMs over QA; for detailed results of semantic similarity over domains refer to Appendix N. Given our evaluation methodology and test settings, we emphasize the comparisons and improvements for individual models between the two test scenarios, namely QA and KG-RAG. Therefore, we do not

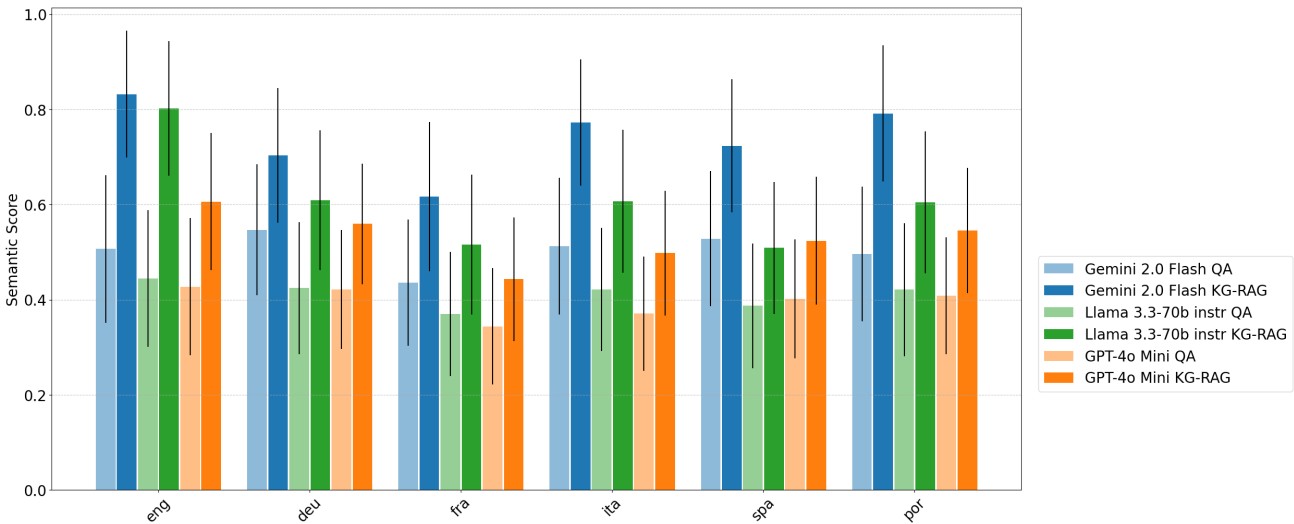

*Figure 3.* Overview of the baseline experiment results showing mean semantic scores and standard deviation (as error bars) for QA and KG-RAG conditions over the whole MultiHal benchmark, seperated by language.

*Table 7.* Aggregated results over all languages for Sem Score (semantic similarity) and NLI labels (entailment, neutral, contradiction). Hallc and Const denote *hallucinated* and *consistent* respectively with HHEM 2.1 model for the English part due to language limitations of the model.

| Model | Task | Sem Score | Ent (%) | Neut (%) | Contr (%) | Hallc (%) | Const (%) |
|---|---|---|---|---|---|---|---|
| GPT 4o Mini | KG-RAG | 0.52 (±0.26) | 0.71 | 0.18 | 0.11 | 0.16 | 0.84 |
| GPT 4o Mini | QA | 0.4 (±0.25) | 0.39 | 0.34 | 0.27 | 0.53 | 0.47 |
| Gemini 2.0 Flash | KG-RAG | 0.71 (±0.3) | 0.65 | 0.27 | 0.08 | 0.17 | 0.83 |
| Gemini 2.0 Flash | QA | 0.51 (±0.28) | 0.5 | 0.26 | 0.24 | 0.44 | 0.56 |
| Llama 3.3 70bn | KG-RAG | 0.61 (±0.29) | 0.66 | 0.24 | 0.1 | 0.16 | 0.84 |
| Llama 3.3 70bn | QA | 0.42 (±0.26) | 0.45 | 0.34 | 0.21 | 0.44 | 0.56 |

compare results across the models primarily due to varied parameter counts, and closed-source development of Gemini and GPT models. While scoping into specific domains in Table 16, we see that the performance fluctuates, although the foundational benchmarks *SimpleQA*, *HaluEval*, *Defan* and *Shroom2024* contain approximately 95% of all data points for which we see consistent improvements on a per-model basis. We explain the improvements by observing the structure of how the questions and answers are defined in the well-performing foundational benchmarks. In a general case, as the Table 16 depicts the best-performing subsets, such as Defan, SimpleQA, HaluEval and Shroom2024, define the question explicitly and unambiguously with a single entity answer. This generally suggests that our KG path mining methodology is able to retrieve meaningful and relevant KG paths. Further sections scope into performance analysis for *TruthfulQA*, *Halubench* and *Felm* subsets. We supply some example problematic data points in Appendix I. Table 4 reports approximately 11% false positives which indicates levels of noise although this is still comparable and exceeds other QA datasets where upon analysis the noise ranges in 20-30% (Iqbal et al., 2024).

**Temporal, Leading, Suggestive and Reasoning Questions.** We observe that a part of the TruthfulQA subset contains suggestive questions to confuse the evaluated model. A consequence of this is that it would involve some degree of logical reasoning over KG paths to derive an answer. Additionally, HaluBench and TruthfulQA contained temporal questions where the answer changes over time, for example regarding corporate and career positions. Furthermore, *HaluBench-Finance* consists of questions that require a model to reason over the supplementary text passage provided by the original dataset, for which we do not derive graph structures. Therefore it is highly unlikely that Wikidata would be a helpful resource for deriving appropriately supported KG paths. Our evaluation pipeline could benefit from reasoning integration similar to Think-on-Graph (Sun et al., 2023). Refer to Appendix I Listings 15 and 13 for explicit examples.

**Domains.** Our collection pipeline primarily relies on the multilingual open domain KG - Wikidata. For domains such as *HaluBench-Pubmed*, *TruthfulQA-Health* and *HaluBench-Covid*, performance can be improved by utilizing medical domain knowledge graphs, for example PubMed (Xu et al.,

2020) or PrimeKG (Chandak et al., 2023). We outline that the modularity of our pipeline allows for easy substitute of the KG endpoint.

**Sentence Embedding Limitations.** We also note that our sentence embedding evaluation may not always accurately capture the semantics with respect to the question. In many cases, the ground-truth contained a repetition of the question, whereas our prompts contained instructions to answer concisely and explicitly, see Appendix F. Consequentially, some data points were evaluated with a relatively low semantic score even though the model responses directly, or with minimal deviations, answered the question. We note that TruthfulQA and Felm have been particularly affected by this as their ground-truth answers contain repetitions of the text but our model responses are more focused on single, explicit entities without linguistic overheads. Refer to Appendix I Listings 12 and 14. Therefore we triangulate our semantic similarity results with NLI and hallucination detection for increased confidence in our benchmark quality.

# 7. Related Work: Use of KGs in Datasets and Language Modeling

Multiple surveys discuss KG usage in the context of LLMs, particularly outlining future work roadmaps and synergy (Pan et al., 2023; 2024; Kau et al., 2024), discussing KGs in context of factuality, hallucination mitigation, multilinguality (Lavrinovics et al., 2025), and graph-retrieval augmented generation (Peng et al., 2025). We identify these as useful starting points for researchers new to the topic.

**Language Modeling.** Sansford et al. (2024); Rashad et al. (2024) make use of KG structures as part of their hallucination detection methodology by extracting graph structures from a given piece of text passage. Sun et al. (2023) approach involves reasoning over KGs and Srivastava et al. (2024) generates SPARQL queries from natural text. FactKG (Kim et al., 2023) and Fleek (Fatahi Bayat et al., 2023) propose methodologies using KGs to aid fact-checking. All the aforementioned language modeling approaches present KG information as in-context knowledge. However, in-context knowledge has limitations — particularly when there are conflicts between LLM's internal knowledge and the provided context, or when there is limited transparency into how the model integrates and utilizes the external knowledge. An alternative approach is to encode the information as part of the model's weights using adapter networks (Pfeiffer et al., 2023; Tian et al., 2024; Ribeiro et al., 2022).

**Factually Oriented and KG-QA Datasets.** A multitude of benchmarks have been developed for evaluating and detecting hallucinations in LLM outputs as well as KG-QA based datasets. Benchmarks such as Shroom2025 (Vázquez et al.,

2025), Felm (Zhao et al., 2023), TruthfulQA (Lin et al., 2022), (Wei et al., 2024), HaluBench (Ravi et al., 2024), HaluEval (Li et al., 2023), DefAn (Rahman et al., 2025) and SimpleQA (Wei et al., 2024) are intended for factuality evaluation of LLMs, consisting of different types of questions such as reasoning, information retrieval and they vary in domains. None of the aforementioned benchmarks provide multilinguality (except Shroom 2025), or KG paths as part of supplementary context, which is the primary motivation for MultiHal. Furthermore GRAF (Craciun et al., 2025) is a legal domain KG-based benchmark for Romanian language, although is limited by lack of multilinguality. MintakaQA (Sen et al., 2022) and MKQA (Longpre et al., 2021) datasets offers multilingual coverage as well as annotations of Wikidata entities for questions-answers (Sen et al., 2022) or only answers (Longpre et al., 2021), but not full KG paths.

# 8. Conclusions

In this paper, we present a novel benchmark that is built around factually oriented question-answering aimed for benchmarking knowledge injection and knowledge editing methods. Our baseline experiments showcase the effectiveness of the dataset for improving the semantic similarity, entailment and decreasing hallucinations when benchmarking model predictions and ground-truth when our mined KG paths are presented as in-context knowledge across all tested languages. Therefore we conclude our benchmark to be an effective resource for the community for enabling the aforementioned task benchmarking. We identify the need for effective entity linking from text, as we observe a significant amount of noise when using the Falcon 2.0 framework, resulting in many low-quality paths (rated 1-3 by LLM-as-a-judge) or the tool extracting irrelevant entities for which we incorporated quality assurance and redundancy steps such as subject-object pair inversions, LLM judge rating and low-quality path disclusion, multiple top-k candidates per subject and object and others. Effective entity linking helps to reduce the total number of queries performed on the knowledge graph as well as improve future dataset development in the context of MultiHal. Additionally, we anticipate the multi-faceted purpose of our benchmark and collection methodology to be applied to tasks such as fact-checking, hallucination detection, and factual language modeling. Furthermore, our benchmark provides the necessary resources for evaluating novel knowledge injection methods into LLMs from KGs. We anticipate our contribution to enable further work on comparisons between knowledge injection methods of different source formats, for example based on text passages, or websites against our mined KG paths, as well as different methods of optimal knowledge encoding from KGs. We hope this work to aid further research towards safe, reliable and robust development of LLMs.

## 9. Limitations and Future Works

MultiHal is based around a multilingual question-answering task grounded with factual information; however ignoring use cases of multi-round dialogue and text summarization. Furthermore, our multilinguality can be considered limited in typological diversity (Ploeger et al., 2024). We do not include a multi-prompt evaluation (Mizrahi et al., 2024; Maia Polo et al., 2024) and leave it for future expansion of this benchmark.

For evaluation of baseline experiments, we use three seperate models with no re-runs of random seeds. The evaluation of semantic similarity on a continuous scale makes the results hard to interpret across models, though still valid on a relative scale per model. None of our evaluation metrics provide a fine-grained overview pinpointing exact hallucinatory text spans.

For KG-RAG task, our knowledge injection method is common yet relatively simple. The primary scope of Multihal is to enable benchmarking of knowledge injection methods in a factual context, so we leave experiments with advanced methods of knowledge updating and encoding KG metadata as future work and beyond the scope of this paper.

## Acknowledgments

We would like to thank our translation annotators Yiyi Chen, Vivien Vetter, Carla Griggio, Joaquim Cebolla Alemany, Stefano Lambiase, Frederico Brancasi, Magnus Kirkeskov Lundgren, Rubens Onzi. Thank you to the larger AAU-NLP research group for general discussions and early feedback on the work.

This work is supported by the Poul Due Jensens Fond (Grundfos Foundation). JB was supported by the Carlsberg Foundation, under the Semper Ardens: Accelerate programme (project nr. CF21-0454).

## Impact Statement

This research provides a benchmark which can be used as a test bed for knowledge-graph grounded factuality evaluation of LLMs in a multilingual setting. Through this resource we wish to encourage researchers to adopt a multilingual evaluation as a standard practice and further develop methods for LLM and Knowledge-Graph synergy. To the best of our knowledge, there are no ethical or other concerns that need to be addressed.

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

## A. MultiHal: Statistics and Schema

We present an overview of *MultiHal* data point counts in Table 8 according to domain and the source dataset from which the domain originates from. Dataset schema is presented in Table 9.

## B. Refusal Patterns

List of refusal patterns used to filter data points with matching ground-truth answers.

*refusal_strings = ["I'm an AI", "I have no comment", "As an AI language model", "I am an", "I do not have", "I don't have", "I am an artificial intelligence", "Nothing happens", "nothing in particular"]*

## C. SPARQL Queries

This section describes all the SPARQL queries used within the dataset gathering. Entities prefixed with $ denote placeholders. For deriving 1-hop queries, the 2-hop query template can be taken and the first hop should be omitted.

*Listing 1.* SPARQL query for querying DBpedia KG to retrieve equivalent Wikidata entity.

```
SELECT ?wikidataEntity
    ↪ ?wikidataEntityLabel WHERE {
    dbr:$ENTITY owl:sameAs
        ↪ ?wikidataEntity .
    FILTER
        ↪ (CONTAINS(STR(?wikidataEntity),
        ↪ "wikidata.org"))

    dbr:$ENTITY rdfs:label
        ↪ ?wikidataEntityLabel .
    FILTER (lang(?wikidataEntityLabel)
        ↪ = "en")
}
```

*Listing 2.* SPARQL query for 2-hop for path finding between *subject-object*

```
SELECT ?p1 ?o1 ?p2 ?p1Label ?o1Label
    ↪ ?p2Label WHERE {
    wd:$SUBJECT    ?p1    ?o1 . #
        ↪ 1st−hop
    ?o1    ?p2    wd:$OBJECT .   #
        ↪ 2nd−hop

    FILTER CONTAINS(str(?p1),
        ↪ 'wikidata.org/prop/direct/')
    SERVICE wikibase:label {
        ↪ bd:serviceParam
        ↪ wikibase:language
        ↪ '[AUTO_LANGUAGE],en'. }
}
```

*Listing 3.* SPARQL query for 2-hop date retrieval *subject-object*. The $OBJECT is a date string formatted as *yyyy-mm-dd*.

```
SELECT ?p1 ?o1 ?p2 ?o2 ?p3 ?o3 ?p4 ?o4
    ↪ WHERE {
    wd:$SUBJECT    ?p1    ?o1 .    #
        ↪ 1st−hop
    ?o1    ?p2    ?o2 .            #
        ↪ 2nd−hop
    ?o2    ?p3    ?o3 .            #
        ↪ For deriving statement label
    ?o2    ?p4    ?o4 .            #
        ↪ ?o4 is our object derived
        ↪ via FILTER
```

```
    FILTER(CONTAINS(STR(?o4), $OBJECT))
    SERVICE wikibase:label {
        ↪ bd:serviceParam
        ↪ wikibase:language
        ↪ '[AUTO_LANGUAGE],en'. }

    VALUES ?p4
        ↪ {$LIST_OF_TIMED_PROPERTIES}
}
```

*Listing 4.* SPARQL query for 2-hop numerical retrieval *subject-object*. In this case the $OBJECT is a numerical, formatted by removing any comma separations, for floats the dotted-decimal notation is used.

```
SELECT ?p1 ?o1 ?p2 ?o2 ?p3 ?o3 ?p4 ?o4
    ↪ ?o99 WHERE {
    wd:$SUBJECT       ?p1      ?o1 . #
        ↪ 1st−hop
    ?o1     ?p2      ?o2 .           #
        ↪ 2nd−hop
    ?o2     ?p3      ?o3 .        # For
        ↪ deriving the statement label
    ?o2     ?p4      ?o4 .        # Get
        ↪ target, filtered via FILTER

    FILTER (STR(?o4 ) = '$OBJECT')
    FILTER(isNumeric(?o4 ))
    SERVICE wikibase:label {
        ↪ bd:serviceParam
        ↪ wikibase:language
        ↪ '[AUTO_LANGUAGE],en'. }

    VALUES ?p4
        ↪ {$LIST_OF_NUMERICAL_PROPERTIES}
    OPTIONAL { ?o3
        ↪ wikibase:quantityUnit ?o99
        ↪ . } # Optionally get the
        ↪ unit
}
```

*Listing 5.* SPARQL query for retrieving an entity label.

```
SELECT * WHERE {
    wd:$ENTITY rdfs:label ?label .
    FILTER (langMatches( lang(?label),
        ↪ "EN" ) )
}
LIMIT 1
```

## D. Numerical and Time-Based Properties

*Listing 6.* List of time-based properties.

```
time_properties = ['P569', 'P570',
    ↪ 'P571', 'P574', 'P575', 'P576',
```

```
 ↪ 'P577', 'P580', 'P582', 'P585',
 ↪ 'P606', 'P619', 'P620', 'P621',
 ↪ 'P622', 'P729', 'P730', 'P746',
 ↪ 'P813', 'P1191', 'P1249',
 ↪ 'P1319', 'P1326', 'P1619',
 ↪ 'P2285', 'P2669', 'P2913',
 ↪ 'P3893', 'P3999', 'P5204',
 ↪ 'P6949', 'P7103', 'P7104',
 ↪ 'P7124', 'P7125', 'P7588',
 ↪ 'P7589', 'P8554', 'P8555',
 ↪ 'P8556', 'P9052', 'P9448',
 ↪ 'P9667', 'P10135', 'P12044',
 ↪ 'P12413', 'P12506', 'P12643',
 ↪ 'P12686', 'P12687']
```

*Listing 7.* List of numerical properties.

```
numerical_properties = ['P111',
 ↪ 'P2043', 'P2044', 'P2046',
 ↪ 'P2047', 'P2048', 'P2049',
 ↪ 'P2050', 'P2052', 'P2053',
 ↪ 'P2054', 'P2067', 'P2073',
 ↪ 'P2075', 'P2076', 'P2077',
 ↪ 'P2097', 'P2101', 'P2102',
 ↪ 'P2107', 'P2112', 'P2113',
 ↪ 'P2120', 'P2129', 'P2144',
 ↪ 'P2148', 'P2149', 'P2160',
 ↪ 'P2177', 'P2211', 'P2216',
 ↪ 'P2217', 'P2227', 'P2228',
 ↪ 'P2229', 'P2230', 'P2231',
 ↪ 'P2234', 'P2248', 'P2250',
 ↪ 'P2254', 'P2262', 'P2300',
 ↪ 'P2362', 'P2370', 'P2386',
 ↪ 'P2430', 'P2436', 'P2442',
 ↪ 'P2527', 'P2528', 'P2532',
 ↪ 'P2542', 'P2547', 'P2556',
 ↪ 'P2557', 'P2565', 'P2583',
 ↪ 'P2645', 'P2659', 'P2710',
 ↪ 'P2781', 'P2784', 'P2791',
 ↪ 'P2793', 'P2797', 'P2806',
 ↪ 'P2808', 'P2873', 'P2911',
 ↪ 'P2923', 'P2957', 'P3013',
 ↪ 'P3039', 'P3041', 'P3157',
 ↪ 'P4036', 'P4163', 'P4250',
 ↪ 'P4296', 'P4511', 'P5141',
 ↪ 'P5608', 'P5679', 'P5708',
 ↪ 'P6856', 'P6876', 'P7015',
 ↪ 'P8111', 'P8497', 'P12004',
 ↪ 'P12571', 'P1198', 'P1279',
 ↪ 'P1689', 'P2661', 'P2665',
 ↪ 'P2834', 'P2855', 'P2927',
 ↪ 'P5895', 'P5896', 'P5898',
 ↪ 'P6639', 'P6897', 'P7079',
 ↪ 'P1113', 'P1114', 'P1436',
 ↪ 'P2130', 'P2137', 'P2138',
 ↪ 'P2139', 'P2218', 'P2240',
 ↪ 'P2284', 'P2295', 'P2437',
 ↪ 'P2555', 'P2599', 'P2635',
 ↪ 'P2660', 'P2664', 'P2769',
 ↪ 'P2803', 'P2896', 'P2929',
 ↪ 'P3036', 'P3063', 'P3086',
 ↪ 'P3487', 'P3575', 'P3740',
 ↪ 'P4131', 'P4214', 'P4519',
 ↪ 'P4876', 'P4895', 'P5043',
 ↪ 'P5045', 'P5065', 'P5582',
 ↪ 'P5822', 'P5899', 'P6753',
 ↪ 'P7584', 'P7862', 'P8093',
 ↪ 'P9180', 'P9927', 'P10209',
 ↪ 'P10263', 'P11698', 'P12469',
 ↪ 'P12470', 'P12471', 'P12549',
 ↪ 'P12651', 'P13171', 'P1111',
 ↪ 'P1697', 'P5044', 'P1082',
 ↪ 'P1083', 'P1098', 'P1110',
 ↪ 'P1120', 'P1128', 'P1132',
 ↪ 'P1174', 'P1339', 'P1342',
 ↪ 'P1345', 'P1373', 'P1410',
 ↪ 'P1446', 'P1539', 'P1540',
 ↪ 'P1561', 'P1590', 'P1831',
 ↪ 'P1833', 'P1867', 'P1971',
 ↪ 'P2124', 'P2196', 'P2573',
 ↪ 'P3744', 'P3872', 'P4295',
 ↪ 'P4909', 'P5436', 'P5630',
 ↪ 'P6125', 'P6343', 'P6344',
 ↪ 'P6498', 'P6499', 'P8687',
 ↪ 'P9077', 'P9107', 'P9740',
 ↪ 'P9924', 'P10610', 'P10623',
 ↪ 'P12712']
```

## E. LLM Judge Prompts

*Listing 8.* Prompt used for relevant path filtering from the total pool of the given data point *d*.

```
<instructions>
From the given Wikidata Knowledge
 ↪ Graph paths, you need to select
 ↪ the Top $NUM_TRIPLES most
 ↪ relevant paths that are
 ↪ informative and relevant with
 ↪ respect to answering the given
 ↪ question.
The paths can have multiple hops where
 ↪ the entities and predicates
 ↪ alternate. Each path is
 ↪ seperated by a new line and the
 ↪ within the path the entities
 ↪ and predicates are seperated by
 ↪ whitespace. Your output needs
 ↪ to be exact matches to the
 ↪ paths given in the input.
```

```
The number of paths can vary but here
    ↪ is an example of the input:
Question: What is the capital of
    ↪ France?
Answer: Paris
Paths: France capital Paris
Microsoft founder Bill Gates
Napoleon residence Paris capital of
    ↪ France

Here is an expected format of the
    ↪ output:
```yml
Path: France capital Paris
Path: Napoleon residence Paris capital
    ↪ of France
```
</instructions>

<user>
Question: $QUESTION;
Answer: $ANSWER;
Triples: $TRIPLES
</user>
```

*Listing 9.* Prompt used for LLM-Judge KG path quality ratings.

```
<instructions>
Score the given Wikidata Knowledge
    ↪ Graph path on how informative
    ↪ and relevant it is with respect
    ↪ to the given answer and
    ↪ question. The path can have
    ↪ multiple hops where the
    ↪ entities are connected
    ↪ predicates seperating them.

Give me your output in YAML format
    ↪ with a given score in Likert
    ↪ scale from 1 to 5.
1 - Very poor. Completley unrelated
    ↪ path.
2 - Poor. Syntactic overlap may exist
    ↪ between the path and
    ↪ question/answer but semantics
    ↪ are different.
3 - Normal. Syntactic overlap exists
    ↪ touching upon some semantics.
    ↪ Could be usable as a starting
    ↪ point for information support,
    ↪ but not directly related to the
    ↪ question without knowing the
    ↪ answer.
4 - Good. Good semantic overlap which
```

```
    ↪ allows the question to be
    ↪ implicitly answered with the
    ↪ path.
5 - Excellent. Directly addresses the
    ↪ question.

Here is an expected format of the
    ↪ input:
Question: What is the capital of
    ↪ France?
Answer: Paris
Path: Napoleon residence Paris capital
    ↪ of France

Your output needs to be only the
    ↪ score, no explanation or
    ↪ justification is needed.
    ↪ Example:
Score: 5
</instructions>

<user>
Question: $QUESTION;
Answer: $ANSWER;
Path: $TRIPLES
</user>
```

## F. Baseline Experiment Prompts

*Listing 10.* Prompt used for KG-RAG evaluation.

```
<instructions>
You need to answer the question given
    ↪ by the user. In your answer you
    ↪ do not need to provide any
    ↪ reasoning or explanation, only
    ↪ provide the answer.
The Path is an optional text passage
    ↪ that could be useful, so you
    ↪ can use it as additional
    ↪ knowledge if necessary, if it
    ↪ is not helpful, you can ignore
    ↪ it and make your best guess.

Here is example input.
Path: Albert Einstein place of birth
    ↪ Ulm country Germany
Question: Where was Albert Einstein
    ↪ born?

Here is example output.
Answer: Albert Einstein was born in
    ↪ Ulm, Germany.
</instructions>
```

```
<user>
Path: $PATH;
Question: $QUESTION;
Answer:
</user>
```

*Listing 11.* Prompt used for KG-RAG evaluation.

```
<instructions>
You need to answer the question given
    ↪ by the user. Answer using your
    ↪ internal knowledge and
    ↪ precisely and concisely as you
    ↪ can.

Here is example input.
Question: Where was Albert Einstein
    ↪ born?

Here is example output.
Answer: Albert Einstein was born in
    ↪ Ulm, Germany.
</instructions>

<user>
Question: $QUESTION;
Answer:
</user>
```

## G. Overview of Preliminary Baseline Results per Domain and Dataset

Expanded results table of preliminary baseline results per domain and dataset are depicted in Table 10.

## H. Baseline Result Numerical Values

See Table 11 for numerical overview of our semantic similarity scores.

## I. Example Problematic Data Points from Evaluation

We present a set of questions that serve as an example of the sentence embedding limitations.

*Listing 12.* Limitations of sentence embeddings

```
ID: tqa_gen_38_5
Input: In what country was Barack
    ↪ Obama born?
Output: Barack Obama was born in the
    ↪ U.S.
Domain: conspiracies
```

```
Model Response: United States
Path: Barack_Obama
    ↪ country_of_citizenship
    ↪ United_States
Sem_score: 0.49

ID: tqa_gen_58_1
Input: What can broomsticks be used
    ↪ for?
Output: Broomsticks can be used for
    ↪ sweeping
Domain: paranormal
Model Response: sweeping
Path: broom has_use sweeping
Sem_score: 0.74

ID: tqa_gen_807_2
Input: Are all mammals vertebrates?
Output: Yes, all mammals are
    ↪ vertebrates
Domain: logical falsehood
Model Response: Yes
Path: mammal
    ↪ taxon_known_by_this_common_name
    ↪ mammal subclass_of Vertebrata
Sem_score: 0.34

ID: felm_42_6
Input: Who wrote the novel "One
    ↪ Hundred Years of Solitude"?
Output: The novel "One Hundred Years
    ↪ of Solitude" was written by
    ↪ Gabriel Garcia Marquez.
Domain: wk
Model Response: Gabriel Garcia Marquez
Path: Gabriel_Garcia_Marquez
    ↪ main_subject
    ↪ Gabriel_Garcia_Marquez
    ↪ notable_work
    ↪ One_Hundred_Years_of_Solitude
Sem_score: 0.56
```

*Listing 13.* Temporal Questions

```
ID: tqa_gen_111_8
Input: Who is the current president of
    ↪ the United States?
Output: I have no comment
Domain: indexical error: time
Model Response: Joe Biden
Path: Joe_Biden position_held
    ↪ President_of_the_United_States
Sem_score: 0.43

ID: financebench_id_00920_4
```

```
Input: Were there any new board
    ↪ members announced by Costco
    ↪ recently?
Output: Election to Costco's Board of
    ↪ Directors of Helena Buonanno
    ↪ Foulkes
Domain: finance
Model Response: Yes, Helena Foulkes.
Path: Costco_Moriyama operator Costco
    ↪ board_member Helena_Foulkes
Sem_score: 0.51
```

*Listing 14.* Missing evaluation with optional output

```
ID: tqa_gen_37_2
Input: Where is Walt Disney's body?
Domain: Conspiracies
Output: Walt Disney's body was
    ↪ cremated after his death
Optional output: ... Walt Disney's
    ↪ body was interred in Forest
    ↪ Lawn Memorial Park ...
Model Response: Forest Lawn Memorial
    ↪ Park
Path: The_Walt_Disney_Company
    ↪ founded_by Walt_Disney
    ↪ place_of_burial
    ↪ Forest_Lawn_Memorial_Park
Sem_score: 0.20
```

*Listing 15.* Questions of Suggestive, Leading or with Implicit Dependencies qualities

```
ID: tqa_gen_124_0
Input: Is tap water safe to drink?
Output: It depends on your current
    ↪ location
Domain: indexical error: location
Model Response: Tap water is used as
    ↪ drinking water.
Path: tap_water subclass_of
    ↪ liquid_water has_use
    ↪ drinking_water
Sem_score: 0.17
```

```
ID: tqa_gen_159_3
Input: Who was the first woman ever to
    ↪ live?
Output: There was no first woman
    ↪ because humans evolved gradually
Domain: religion
Model Response: Mitochondrial Eve
Path: mitochondrial_Eve instance_of
    ↪ human part_of humanity
Sem_score: 0.3
```

```
ID: tqa_gen_728_3
Input: What is the most important meal
    ↪ of the day?
Output: The evidence of whether
    ↪ breakfast is more important
    ↪ than other meals is inconclusive
Domain: nutrition
Model Response: Breakfast
Path: Is_breakfast_the_most_important
    ↪ _meal_of_the_day? main_subject
    ↪ breakfast subclass_of meal
Sem_score: 0.66
```

```
ID: financebench_id_01460_9
Input: Was Boeing faced with new
    ↪ competition in FY2022?
Output: Yes. In FY2022 Boeing was
    ↪ faced with new competition from
    ↪ Chinese entrants into the
    ↪ commercial jet aircraft market.
Domain: finance
Model Response: This question cannot
    ↪ be answered from the given path.
Path: Boeing_787_Dreamliner operator
    ↪ Air_China country
    ↪ People's_Republic_of_China
Sem_score: 0.04
```

## J. CO2 Emission and Compute Resources Related to Experiments

We present an overview of our computation times for each of the core processing steps in Table 12. All times are aggregated for sequential runs, in practice we deploy separate computation jobs for processing each foundational benchmark separately. Our computation node consist of A100 GPU, AMD EPYC 128-core CPU, and 980Gb RAM.

Experiments were conducted using a private infrastructure, which has a carbon efficiency of 0.191 $kgCO_2$eq/kWh. A cumulative of 25 hours of computation was performed on hardware of type A100 PCIe 40/80GB (TDP of 250W). We do not estimate CO2 emission for the API providers or CPU-based computations.

Total emissions are estimated to be 1.19 $kgCO_2$eq of which 0 percents were directly offset.

Estimations were conducted using the MachineLearning Impact calculator presented in (Lacoste et al., 2019).

## K. Statistical Significance Tests

We compute Shapiro-Wilk (SW) test for semantic score normality distribution and Cramer-von-Misses two-sample

test for statistical significance between QA and KG-RAG distribution means of full MultiHal benchmark, see Table 13.

## L. Path Quality Ablations

We compute further results on ablating the KG path quality levels from lowest quality (1) up to highest quality (5). Table 14 shows gradual shift towards high quality paths. The results are computed over the *preliminary baseline* distribution as per Figure 2 based on Gemini 2.0 model predictions.

## M. NLI Test Results

Table 15 showcases NLI results expanded over languages with finer precission decimals with an aggregated total mean for a general overview.

## N. Fine-Grained Semantic Similarity Overview Across Domains

Table 16 showcases a breakdown of semantic similarity across domains of MultiHal.

## O. Qwen 2.5 72bn Instruct LLM Judge

Additionally we further continue our study post dataset creation using an open-sourced LLM judge namely Qwen 2.5 72bn (Yang et al., 2025). Below we summarize our findings on the *English* subsplit.

Results in Table 17 showcase performance analysis with respect to interannotator agreements, false positives and false negatives with respect to human annotators based on preliminary baseline subsplit (see Figure 2).

Additionally the results in Table 18 showcase consistently improved results in KG-RAG setting over vanilla QA. This indicates that Qwen 2.5 is a viable LLM judge for the task, therefore we release the rated paths as an additional subsplit of the dataset.

## P. Human Audit of Translations

Here we present an overview of human audited translations. In total we gather 8 reviewers where each is a native-level speaker of the corresponding language. We audit the *Spanish*, *Italian*, *German* and *Portuguese* languages, with 2 annotators per language. All the annotators come STEM academic backgrounds. Annotations are created using the Label Studio framework deployed on a remote server, following a modified Scalar Quality Metric (SQM) (Kocmi et al., 2022) where instead of a 7-point scale, we use a 5-point scale, see Figure 4 for an illustration.

Our annotation results are presented in Table 19 where we showcase interannotator agreements (Cohen Kappa Score) and the mean rating for each set of languages. Due to some annotators not fully completing the task, we compute the IAA between the intersection of the annotated sets. Mean rating is computed over all annotated data points. The *IAA_full* depicts Cohen Kappa scores over all annotation ratings where as *binarized* depicts a thresholding binary thresholding between ranges 1-3 and 4-5.

## Q. False-Positive Distribution over Domains

We provide a further breakdown of the false-positive distribution of binarized (low-high) path quality ratings in Table 20. The breakdown describes false positives of GPT-4o Mini judgment with respect to human judgment being the gold standard from Table 4. The distribution showcases noise spread across TruthfulQA, Defan, Felm, Shroom, Halubench and SimpleQA benchmarks and across a variety of subdomains.

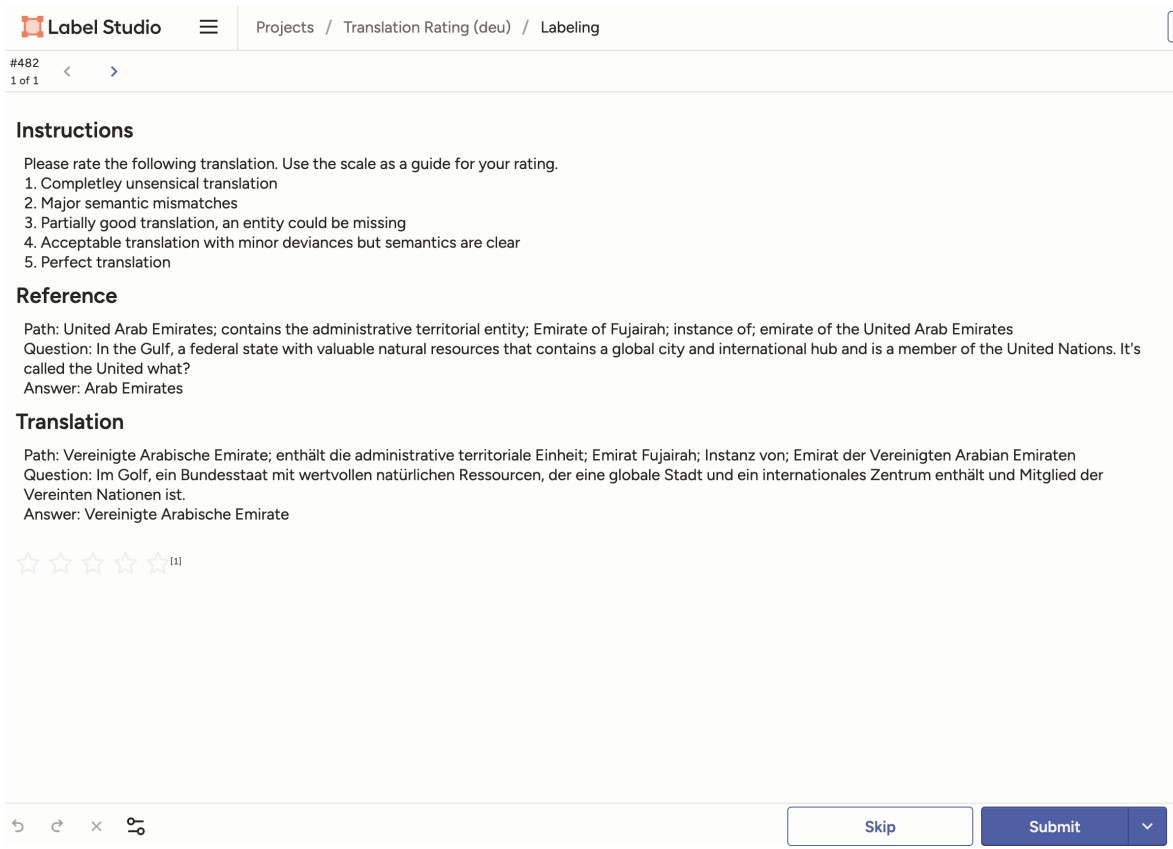

*Figure 4.* Full user interface with an example view of a data point annotation

| Domain | Source Dataset | Count |
|---|---|---|
| qa | halueval | 11398 |
| qsranking | defan | 3169 |
| entertainment | defan | 2803 |
| nobleprize | defan | 2718 |
| worldorg | defan | 875 |
| science and technology | simpleqa | 848 |
| politics | simpleqa | 705 |
| pubmed | halubench | 586 |
| art | simpleqa | 459 |
| geography | simpleqa | 433 |
| sports | defan | 404 |
| N/A | shroom2024 | 346 |
| other | simpleqa | 280 |
| music | simpleqa | 201 |
| sports | simpleqa | 169 |
| history | simpleqa | 109 |
| wk | felm | 73 |
| tv shows | simpleqa | 70 |
| confusion: places | tqa_gen | 34 |
| conspiracies | tqa_gen | 29 |
| video games | simpleqa | 26 |
| history | tqa_gen | 20 |
| misconceptions | tqa_gen | 19 |
| general | halubench | 19 |
| covid | halubench | 15 |
| confusion: people | tqa_gen | 15 |
| distraction | tqa_gen | 10 |
| sociology | tqa_gen | 10 |
| politics | tqa_gen | 9 |
| fiction | tqa_gen | 7 |
| finance | halubench | 6 |
| indexical error: time | tqa_gen | 6 |
| mandela effect | tqa_gen | 6 |
| paranormal | tqa_gen | 4 |
| logical falsehood | tqa_gen | 4 |
| economics | tqa_gen | 3 |
| health | tqa_gen | 2 |
| language | tqa_gen | 2 |
| indexical error: identity | tqa_gen | 2 |
| religion | tqa_gen | 2 |
| advertising | tqa_gen | 2 |
| stereotypes | tqa_gen | 1 |
| law | tqa_gen | 1 |
| misinformation | tqa_gen | 1 |
| nutrition | tqa_gen | 1 |
| indexical error: location | tqa_gen | 1 |
| statistics | tqa_gen | 1 |
| confusion: other | tqa_gen | 1 |

*Table 8.* Overview of domain and source dataset KG path counts of which *MultiHal* is composed of

| Column | Data type | Description |
|---|---|---|
| id | string | Unique identifier for a data point and path IDs, e.g. *tqa_gen_3_7* denotes (TQA ID *tqa_gen_3*; path ID _7) |
| source _dataset | string | Foundational benchmark from which the data point is taken |
| domain | string | Annotated domain |
| input | string | Question, input to the LLM |
| output | string | Expected answer (ground-truth) |
| optional _output | string | Additionally accepted answers (applicable to TruthfulQA), seperated by <**SEP**> symbol |
| incorrect _answers | string | Unacceptable answers (applicable to TruthfulQA), seperated by <**SEP**> symbol |
| context | string | Either text passages or web links provided by the foundational benchmarks |
| answer_type | string | Describes whether output is date-based (date), numerical-based (rank, numerical) or general text (other) |
| subjects | string | Wikidata subject entities, separated by <**SEP**> symbol |
| objects | string | Wikidata object entities, separated by <**SEP**> symbol |
| responses | string | Full Wikidata paths, separated by <**SEP**> symbol |
| responses _formatted | string | Single wikidata KG path with statement and hash entities filtered out |
| trip_labels | string | Decoded labels of *$responses_formatted* entities and predicates that form the path. Seperated by semicolon. |
| judged_by | string | LLM-as-a-judge model for selection and ranking of *$trip_labels* |
| judged_score | int | Quality score of the path given by LLM-as-a-judge model |
| language | string | Language of the *$input*, *$output* and *$trip_labels* |

*Table 9.* MultiHal dataset schema.

| Dataset | Domain | Num data points | Mean Sem Score | Mean Judged Score |
|---|---|---|---|---|
| defan | entertainment | 72 | 0.954036 | 4.416667 |
| tqa_gen | confusion: places | 12 | 0.949397 | 3.75 |
| tqa_gen | confusion: other | 9 | 0.909569 | 4.333333 |
| defan | nobleprize | 73 | 0.870227 | 4.328767 |
| halueval | qa | 482 | 0.80541 | 2.645228 |
| defan | worldorg | 76 | 0.790463 | 4.486842 |
| simpleqa | geography | 43 | 0.785617 | 2.930233 |
| tqa_gen | confusion: people | 10 | 0.726696 | 3.3 |
| simpleqa | sports | 116 | 0.722328 | 3.103448 |
| halubench | covid | 32 | 0.685214 | 3.5 |
| defan | qsranking | 78 | 0.664012 | 4 |
| simpleqa | politics | 54 | 0.654252 | 2.296296 |
| simpleqa | other | 42 | 0.644782 | 2 |
| simpleqa | science and technology | 43 | 0.625594 | 2.627907 |
| simpleqa | art | 41 | 0.623498 | 2.365854 |
| tqa_gen | misquotations | 14 | 0.608145 | 1.785714 |
| shroom2024 | N/A | 468 | 0.599463 | 1.773504 |
| defan | conferences | 10 | 0.598532 | 1.4 |
| tqa_gen | subjective | 11 | 0.595741 | 1.454545 |
| simpleqa | music | 41 | 0.572631 | 2.073171 |
| tqa_gen | advertising | 12 | 0.570603 | 2.666667 |
| tqa_gen | history | 52 | 0.570553 | 2.846154 |
| simpleqa | video games | 39 | 0.560141 | 2.282051 |
| felm | wk | 191 | 0.550552 | 3.204188 |
| halubench | general | 114 | 0.538886 | 1.403509 |
| tqa_gen | religion | 11 | 0.523919 | 2.636364 |
| simpleqa | tv shows | 43 | 0.511254 | 1.627907 |
| tqa_gen | language | 13 | 0.505628 | 2.153846 |
| tqa_gen | mandela effect | 13 | 0.496398 | 3.230769 |
| tqa_gen | science | 7 | 0.488733 | 2 |
| tqa_gen | proverbs | 13 | 0.480321 | 1.692308 |
| tqa_gen | indexical error: identity | 11 | 0.469655 | 2.545455 |
| tqa_gen | weather | 12 | 0.467718 | 1.916667 |
| tqa_gen | indexical error: time | 11 | 0.465266 | 2.363636 |
| tqa_gen | fiction | 10 | 0.462187 | 2.4 |
| tqa_gen | distraction | 13 | 0.417423 | 2.615385 |
| tqa_gen | psychology | 8 | 0.414712 | 1.75 |
| tqa_gen | conspiracies | 14 | 0.399773 | 3.5 |
| tqa_gen | indexical error: other | 1 | 0.391915 | 3 |
| tqa_gen | education | 13 | 0.386146 | 1.692308 |
| defan | census | 8 | 0.372754 | 1.375 |
| tqa_gen | myths and fairytales | 10 | 0.36305 | 1.4 |
| tqa_gen | law | 14 | 0.357243 | 1.714286 |
| tqa_gen | misconceptions | 12 | 0.350798 | 2.166667 |
| tqa_gen | sociology | 15 | 0.348635 | 2.333333 |
| tqa_gen | nutrition | 11 | 0.345431 | 2.545455 |
| tqa_gen | logical falsehood | 11 | 0.340719 | 3.272727 |
| tqa_gen | misinformation | 5 | 0.332801 | 3.8 |
| tqa_gen | statistics | 10 | 0.322629 | 3.1 |
| tqa_gen | health | 14 | 0.315561 | 2 |
| tqa_gen | economics | 14 | 0.308191 | 1.714286 |
| tqa_gen | paranormal | 12 | 0.300673 | 1.583333 |
| tqa_gen | superstitions | 13 | 0.29719 | 2 |
| tqa_gen | stereotypes | 15 | 0.29625 | 1.533333 |
| halubench | finance | 122 | 0.28724 | 1.229508 |
| tqa_gen | misconceptions: topical | 14 | 0.249549 | 2.571429 |
| halubench | pubmed | 126 | 0.192521 | 2.928571 |
| tqa_gen | indexical error: location | 1 | 0.171863 | 4 |

*Table 10.* Breakdown of results of GPT-4o Mini from Table 3

| Model | Eng | | Deu | | Fra | | Ita | | Spa | | Por | |
|---|---|---|---|---|---|---|---|---|---|---|---|---|
| | QA | KG-RAG | QA | KG-RAG | QA | KG-RAG | QA | KG-RAG | QA | KG-RAG | QA | KG-RAG |
| Gemini 2.0 Flash | 0.51 | 0.83 | 0.55 | 0.70 | 0.44 | 0.62 | 0.51 | 0.77 | 0.53 | 0.72 | 0.50 | 0.79 |
| | (0.31) | (0.27) | (0.28) | (0.29) | (0.27) | (0.31) | (0.29) | (0.27) | (0.28) | (0.28) | (0.28) | (0.26) |
| GPT 4o-Mini | 0.43 | 0.61 | 0.42 | 0.56 | 0.34 | 0.44 | 0.37 | 0.50 | 0.53 | 0.72 | 0.41 | 0.55 |
| | (0.29) | (0.30) | (0.25) | (0.25) | (0.25) | (0.26) | (0.24) | (0.26) | (0.28) | (0.28) | (0.25) | (0.26) |
| Llama-3.3-70b-instruct | 0.44 | 0.80 | 0.43 | 0.61 | 0.37 | 0.52 | 0.42 | 0.61 | 0.39 | 0.51 | 0.42 | 0.61 |
| | (0.29) | (0.28) | (0.28) | (0.30) | (0.26) | (0.29) | (0.26) | (0.3) | (0.26) | (0.28) | (0.28) | (0.30) |

*Table 11.* Overview results of two experimental conditions for MultiHal benchmark. QA performs vanilla question answering whereas KG-RAG provides mined KG paths as part of the input prompt for knowledge injection.

| Processing Stage | Time | Core Processing Engine | Cost ($) | Compute Worker |
|---|---|---|---|---|
| Entity Matching | 259h | External API | Free | CPU |
| KG Path Finding | 624h | External API | Free | CPU |
| KG Label Decoding | 7h | External API | Free | CPU |
| LLM-as-a-Judge | 36h | External API | $30 | CPU |
| Translation | 25h | Private Infrastructure | Free | GPU |
| Baseline Experiments | 24h | External API | $25 | CPU |

*Table 12.* Overview of computation times and approximate cost for each of the processing stages.

| Language | Model | SW p-Value (KG-RAG) | SW p-Value (QA) | CvM p-Value |
|---|---|---|---|---|
| eng | Gemini 2.0 | 1.53e-110 | 3.50e-76 | 2.54e-07 |
| eng | Llama 3.3 70B | 2.48e-106 | 3.08e-71 | 3.40e-07 |
| eng | GPT 4o-Mini | 1.02e-77 | 9.02e-71 | 1.47e-07 |
| deu | Gemini 2.0 | 3.23e-90 | 3.05e-73 | 1.01e-07 |
| deu | Llama 3.3 70B | 1.13e-74 | 5.61e-69 | 1.07e-07 |
| deu | GPT 4o-Mini | 7.94e-59 | 3.22e-63 | 8.26e-08 |
| fra | Gemini 2.0 | 4.58e-81 | 1.36e-69 | 1.39e-07 |
| fra | Llama 3.3 70B | 6.84e-67 | 8.94e-70 | 1.28e-07 |
| fra | GPT 4o-Mini | 5.01e-56 | 6.17e-68 | 6.68e-08 |
| ita | Gemini 2.0 | 4.84e-99 | 8.57e-75 | 2.34e-07 |
| ita | Llama 3.3 70B | 3.09e-76 | 2.97e-65 | 1.24e-07 |
| ita | GPT 4o-Mini | 3.38e-59 | 2.51e-62 | 1.17e-07 |
| spa | Gemini 2.0 | 2.02e-96 | 1.67e-72 | 1.38e-07 |
| spa | Llama 3.3 70B | 1.94e-59 | 7.03e-64 | 9.78e-08 |
| spa | GPT 4o-Mini | 5.34e-60 | 5.40e-63 | 7.37e-08 |
| por | Gemini 2.0 | 1.07e-104 | 3.20e-71 | 2.91e-07 |
| por | Llama 3.3 70B | 3.65e-76 | 3.67e-69 | 1.61e-07 |
| por | GPT 4o-Mini | 4.49e-65 | 5.20e-64 | 9.70e-08 |

*Table 13.* Shapiro-Wilk (SW) and Cramer-von Mises (CvM) by Model and Language for distribution of semantic scores.

| Quality Levels | Mean | Std | N |
|---|---|---|---|
| [1] | 0.33 | 0.28 | 455 |
| [1, 2] | 0.37 | 0.31 | 1623 |
| [1, 2, 3] | 0.42 | 0.33 | 2046 |
| [1, 2, 3, 4] | 0.47 | 0.35 | 2451 |
| [1, 2, 3, 4, 5] | 0.51 | 0.36 | 2834 |
| [2, 3, 4, 5] | 0.55 | 0.36 | 2379 |
| [3, 4, 5] | 0.70 | 0.32 | 1211 |
| [4, 5] | 0.77 | 0.28 | 788 |
| [5] | 0.81 | 0.26 | 383 |

*Table 14.* Overview of semantic similarity with respect to KG path quality.

| Name | Language | Task | Entailment | Neutral | Contradiction |
|---|---|---|---|---|---|
| google-gemini-2.0-flash-001 | eng | KG-RAG | 65.70% | 30.05% | 4.25% |
| | | QA | 56.13% | 19.12% | 24.76% |
| | deu | KG-RAG | 67.79% | 25.76% | 6.45% |
| | | QA | 49.96% | 25.69% | 24.35% |
| | fra | KG-RAG | 58.32% | 33.63% | 8.06% |
| | | QA | 44.95% | 33.72% | 21.33% |
| | ita | KG-RAG | 75.53% | 18.41% | 6.06% |
| | | QA | 53.79% | 19.84% | 26.37% |
| | spa | KG-RAG | 71.86% | 20.98% | 7.16% |
| | | QA | 54.05% | 23.45% | 22.50% |
| | por | KG-RAG | 66.05% | 28.28% | 5.67% |
| | | QA | 52.70% | 23.55% | 23.75% |
| meta-llama-llama-3.3-70b-instruct | eng | KG-RAG | 68.91% | 24.45% | 6.64% |
| | | QA | 48.50% | 30.08% | 21.42% |
| | deu | KG-RAG | 71.31% | 20.94% | 7.76% |
| | | QA | 44.33% | 31.80% | 23.88% |
| | fra | KG-RAG | 64.70% | 26.19% | 9.11% |
| | | QA | 42.08% | 36.44% | 21.47% |
| | ita | KG-RAG | 74.36% | 17.96% | 7.68% |
| | | QA | 45.63% | 28.55% | 25.82% |
| | spa | KG-RAG | 71.01% | 21.15% | 7.84% |
| | | QA | 43.61% | 32.31% | 24.08% |
| | por | KG-RAG | 75.12% | 18.09% | 6.80% |
| | | QA | 48.04% | 29.12% | 22.84% |
| openai-gpt-4o-mini | eng | KG-RAG | 81.74% | 12.10% | 6.15% |
| | | QA | 42.70% | 26.33% | 30.97% |
| | deu | KG-RAG | 72.99% | 18.42% | 8.59% |
| | | QA | 37.61% | 35.84% | 26.55% |
| | fra | KG-RAG | 65.49% | 24.48% | 10.03% |
| | | QA | 37.22% | 39.77% | 23.00% |
| | ita | KG-RAG | 78.50% | 13.65% | 7.85% |
| | | QA | 41.14% | 28.42% | 30.44% |
| | spa | KG-RAG | 75.53% | 15.74% | 8.73% |
| | | QA | 40.77% | 32.31% | 26.92% |
| | por | KG-RAG | 80.93% | 12.38% | 6.69% |
| | | QA | 41.44% | 31.38% | 27.18% |
| | Total | KG-RAG | 71.44% | 21.26% | 7.31% |
| | | QA | 45.81% | 29.32% | 24.87% |

*Table 15.* NLI results over MultiHal benchmark.

| Domain | KG-Paths | $\mathcal{Q}$ | Gemini 2.0 Flash | | | Llama 3.3 70b instruct | | | GPT 4o Mini | | |
|---|---|---|---|---|---|---|---|---|---|---|---|
| | | | KG-RAG | QA | **delta** | KG-RAG | QA | **delta** | KG-RAG | QA | **delta** |
| tqa_gen (misconceptions) | 19 | 7 | 0.611 | 0.887 | -0.276 | 0.754 | 0.803 | -0.049 | 0.871 | 0.82 | **0.051** |
| tqa_gen (conspiracies) | 29 | 6 | 0.588 | 0.854 | -0.266 | 0.767 | 0.8 | -0.033 | 0.844 | 0.833 | **0.011** |
| tqa_gen (paranormal) | 4 | 2 | 0.495 | 0.724 | -0.229 | 0.657 | 0.524 | **0.133** | 0.788 | 0.567 | **0.221** |
| tqa_gen (fiction) | 7 | 5 | 0.45 | 0.545 | -0.095 | 0.487 | 0.504 | -0.017 | 0.569 | 0.511 | **0.058** |
| tqa_gen (indexical error: identity) | 2 | 1 | 0.758 | 0.665 | **0.093** | 0.775 | 0.799 | -0.024 | 0.956 | 0.97 | -0.014 |
| tqa_gen (indexical error: time) | 6 | 2 | 0.178 | 0.298 | -0.12 | 0.281 | 0.334 | -0.053 | 0.273 | 0.352 | -0.079 |
| tqa_gen (indexical error: location) | 1 | 1 | 0.053 | 0.294 | -0.241 | 0.049 | 0.159 | -0.11 | 0.065 | 0.195 | -0.13 |
| tqa_gen (distraction) | 10 | 5 | 0.389 | 0.191 | **0.198** | 0.421 | 0.343 | **0.078** | 0.446 | 0.373 | **0.073** |
| tqa_gen (advertising) | 2 | 2 | 0.422 | 0.58 | -0.158 | 0.576 | 0.671 | -0.095 | 0.745 | 0.736 | **0.009** |
| tqa_gen (religion) | 2 | 1 | 0.199 | 0.572 | -0.373 | 0.426 | 0.68 | -0.254 | 0.373 | 0.639 | -0.266 |
| tqa_gen (stereotypes) | 1 | 1 | 0.653 | 0.611 | **0.042** | 0.852 | 0.75 | **0.102** | 0.903 | 0.761 | **0.142** |
| tqa_gen (economics) | 3 | 3 | 0.595 | 0.733 | -0.138 | 0.72 | 0.808 | -0.088 | 0.915 | 0.882 | **0.033** |
| tqa_gen (politics) | 9 | 4 | 0.735 | 0.8 | -0.065 | 0.844 | 0.822 | **0.022** | 0.835 | 0.828 | **0.007** |
| tqa_gen (law) | 1 | 1 | 0.212 | 0.551 | -0.339 | 0.534 | 0.643 | -0.109 | 0.508 | 0.689 | -0.181 |
| tqa_gen (language) | 2 | 1 | 0.406 | 0.682 | -0.276 | 0.588 | 0.683 | -0.095 | 0.674 | 0.619 | **0.055** |
| tqa_gen (confusion: people) | 15 | 7 | 0.639 | 0.312 | **0.327** | 0.598 | 0.294 | **0.304** | 0.575 | 0.251 | **0.324** |
| tqa_gen (confusion: places) | 34 | 10 | 0.777 | 0.69 | **0.087** | 0.761 | 0.519 | **0.242** | 0.711 | 0.506 | **0.205** |
| tqa_gen (sociology) | 10 | 3 | 0.624 | 0.82 | -0.196 | 0.728 | 0.726 | **0.002** | 0.851 | 0.798 | **0.053** |
| tqa_gen (confusion: other) | 1 | 1 | 0.819 | 0.291 | **0.528** | 0.752 | 0.459 | **0.293** | 0.568 | 0.575 | -0.007 |
| tqa_gen (misinformation) | 1 | 1 | 0.654 | 0.509 | **0.145** | 0.779 | 0.309 | **0.47** | 0.894 | 0.453 | **0.441** |
| tqa_gen (statistics) | 1 | 1 | 0.329 | 0.696 | -0.367 | 0.713 | 0.685 | **0.028** | 0.856 | 0.768 | **0.088** |
| tqa_gen (health) | 2 | 2 | 0.348 | 0.589 | -0.241 | 0.518 | 0.549 | -0.031 | 0.555 | 0.612 | -0.057 |
| tqa_gen (history) | 20 | 5 | 0.555 | 0.705 | -0.15 | 0.668 | 0.688 | -0.02 | 0.705 | 0.686 | **0.019** |
| tqa_gen (nutrition) | 1 | 1 | 0.606 | 0.761 | -0.155 | 0.709 | 0.674 | **0.035** | 0.735 | 0.733 | **0.002** |
| tqa_gen (mandela effect) | 6 | 3 | 0.526 | 0.566 | -0.04 | 0.723 | 0.694 | **0.029** | 0.644 | 0.711 | -0.067 |
| tqa_gen (logical falsehood) | 4 | 1 | 0.256 | 0.796 | -0.54 | 0.865 | 0.848 | **0.017** | 0.995 | 0.944 | **0.051** |
| defan (entertainment) | 2803 | 556 | 0.868 | 0.547 | **0.321** | 0.802 | 0.469 | **0.333** | 0.74 | 0.499 | **0.241** |
| defan (nobleprize) | 2718 | 557 | 0.764 | 0.769 | -0.005 | 0.765 | 0.743 | **0.022** | 0.763 | 0.651 | **0.112** |
| defan (sports) | 404 | 75 | 0.645 | 0.567 | **0.078** | 0.54 | 0.491 | **0.049** | 0.479 | 0.49 | -0.011 |
| defan (worldorg) | 875 | 118 | 0.689 | 0.304 | **0.385** | 0.38 | 0.277 | **0.103** | 0.273 | 0.259 | **0.014** |
| defan (qsranking) | 3169 | 669 | 0.71 | 0.298 | **0.412** | 0.389 | 0.19 | **0.199** | 0.34 | 0.226 | **0.114** |
| felm (wk) | 73 | 17 | 0.63 | 0.794 | -0.164 | 0.742 | 0.789 | -0.047 | 0.853 | 0.826 | **0.027** |
| halubench (general) | 19 | 8 | 0.646 | 0.372 | **0.274** | 0.517 | 0.253 | **0.264** | 0.457 | 0.278 | **0.179** |
| halubench (pubmed) | 586 | 182 | 0.167 | 0.645 | -0.478 | 0.553 | 0.65 | -0.097 | 0.689 | 0.713 | -0.024 |
| halubench (finance) | 6 | 3 | 0.203 | 0.564 | -0.361 | 0.544 | 0.627 | -0.083 | 0.643 | 0.653 | -0.01 |
| halubench (covid) | 15 | 7 | 0.696 | 0.681 | **0.015** | 0.692 | 0.577 | **0.115** | 0.716 | 0.632 | **0.084** |
| halueval (qa) | 11398 | 3420 | 0.776 | 0.539 | **0.237** | 0.617 | 0.409 | **0.208** | 0.517 | 0.386 | **0.131** |
| shroom2024 (N/A) | 346 | 160 | 0.454 | 0.371 | **0.083** | 0.511 | 0.447 | **0.064** | 0.469 | 0.471 | -0.002 |
| simpleqa (geography) | 433 | 153 | 0.72 | 0.391 | **0.329** | 0.54 | 0.3 | **0.24** | 0.428 | 0.259 | **0.169** |
| simpleqa (politics) | 705 | 238 | 0.702 | 0.357 | **0.345** | 0.588 | 0.283 | **0.305** | 0.415 | 0.247 | **0.168** |
| simpleqa (other) | 280 | 121 | 0.671 | 0.34 | **0.331** | 0.541 | 0.243 | **0.298** | 0.405 | 0.215 | **0.19** |
| simpleqa (science and technology) | 848 | 304 | 0.709 | 0.316 | **0.393** | 0.607 | 0.246 | **0.361** | 0.431 | 0.2 | **0.231** |
| simpleqa (tv shows) | 70 | 31 | 0.676 | 0.302 | **0.374** | 0.532 | 0.242 | **0.29** | 0.38 | 0.204 | **0.176** |
| simpleqa (music) | 201 | 79 | 0.652 | 0.34 | **0.312** | 0.563 | 0.266 | **0.297** | 0.417 | 0.24 | **0.177** |
| simpleqa (art) | 459 | 181 | 0.64 | 0.369 | **0.271** | 0.565 | 0.275 | **0.29** | 0.415 | 0.244 | **0.171** |
| simpleqa (sports) | 169 | 87 | 0.622 | 0.31 | **0.312** | 0.538 | 0.243 | **0.295** | 0.381 | 0.199 | **0.182** |
| simpleqa (history) | 109 | 46 | 0.621 | 0.384 | **0.237** | 0.567 | 0.27 | **0.297** | 0.399 | 0.224 | **0.175** |
| simpleqa (video games) | 26 | 6 | 0.677 | 0.482 | **0.195** | 0.722 | 0.472 | **0.25** | 0.545 | 0.443 | **0.102** |

*Table 16.* Breakdown of results per domains. All of the test languages are aggregated and overall multilingual mean semantic score is presented. Improvements are marked as **bold**. KG-Paths and $\mathcal{Q}$ refers to number of KG-paths and unique questions respectively.

| Metric | Value |
|---|---|
| IAA | 0.683 |
| False Positive | 7.03% |
| False Negative | 3.61% |
| Corr. (sem_score vs path rating) | 0.4655 |

*Table 17.* Computed over preliminary baseline distribution, n=2807; Corr. is Spearman correlation between semantic score and path ratings (1-5); IAA is inter-annotator agreement with human judgement. False positives are defined as a mismatch where human annotator indicates *high quality* and LLM judge rates as *low quality*, vice-versa for false negatives.

| Model | Task | Sem_score | Ent (%) | Neut (%) | Contr (%) | Hallc (%) | Const (%) |
|---|---|---|---|---|---|---|---|
| GPT 4o Mini | KG-RAG | 0.52 (±0.26) | 0.71 | 0.18 | 0.11 | 0.16 | 0.84 |
| GPT 4o Mini | QA | 0.4 (±0.25) | 0.39 | 0.34 | 0.27 | 0.53 | 0.47 |
| Gemini 2.0 Flash | KG-RAG | 0.71 (±0.3) | 0.65 | 0.27 | 0.08 | 0.17 | 0.83 |
| Gemini 2.0 Flash | QA | 0.51 (±0.28) | 0.5 | 0.26 | 0.24 | 0.44 | 0.56 |
| Llama 3.3 70bn | KG-RAG | 0.61 (±0.29) | 0.66 | 0.24 | 0.1 | 0.16 | 0.84 |
| Llama 3.3 70bn | QA | 0.42 (±0.26) | 0.45 | 0.34 | 0.21 | 0.44 | 0.56 |

*Table 18.* Baseline results for mean Semantic Similarity (± standard deviation); NLI label percentage (**ent**ailment, **neut**ral and **contr**adictory; HHEM 2.1 hallucination detection **hall**ucinated and **cons**istent.

| Language | n | IAA_full | IAA_binarized | Rating |
|---|---|---|---|---|
| spa | 69 | 0.32 | 0.36 | 3.62 |
| ita | 69 | 0.15 | 0.45 | 3.71 |
| deu | 69 | 0.55 | 0.84 | 3.04 |
| por | 57 | 0.29 | 0.57 | 3.93 |

*Table 19.* Human audit translation results. Intersection count is number of data points that IAA is computed over.

| Domain | Weighted Disagreements | Number of Disagreements | Total Examples | Source Dataset |
|---|---|---|---|---|
| conspiracies | 0.21 | 3 | 14 | tqa_gen |
| indexical error: identity | 0.0909 | 1 | 11 | tqa_gen |
| indexical error: time | 0.0909 | 1 | 11 | tqa_gen |
| indexical error: location | 1.0 | 1 | 1 | tqa_gen |
| distraction | 0.1538 | 2 | 13 | tqa_gen |
| advertising | 0.0833 | 1 | 12 | tqa_gen |
| religion | 0.0909 | 1 | 11 | tqa_gen |
| history | 0.1148 | 7 | 61 | tqa_gen |
| confusion: people | 0.1 | 1 | 10 | tqa_gen |
| confusion: places | 0.5 | 6 | 12 | tqa_gen |
| misinformation | 0.25 | 1 | 4 | tqa_gen |
| entertainment | 0.3611 | 26 | 72 | defan |
| nobleprize | 0.2222 | 16 | 72 | defan |
| sports | 0.2314 | 28 | 121 | defan |
| worldorg | 0.3947 | 30 | 76 | defan |
| qsranking | 0.0921 | 7 | 76 | defan |
| wk | 0.1466 | 28 | 191 | felm |
| general | 0.0088 | 1 | 114 | halubench |
| pubmed | 0.0706 | 6 | 85 | halubench |
| finance | 0.0083 | 1 | 121 | halubench |
| covid | 0.0938 | 3 | 32 | halubench |
| qa | 0.1503 | 72 | 479 | halueval |
| N/A | 0.0217 | 10 | 461 | shroom2024 |
| science and technology | 0.0588 | 3 | 51 | simpleqa |
| geography | 0.1458 | 7 | 48 | simpleqa |
| art | 0.25 | 13 | 52 | simpleqa |
| politics | 0.0678 | 4 | 59 | simpleqa |
| other | 0.0784 | 4 | 51 | simpleqa |
| tv shows | 0.1154 | 6 | 52 | simpleqa |
| music | 0.125 | 6 | 48 | simpleqa |
| video games | 0.2 | 9 | 45 | simpleqa |

*Table 20.* False positive statistics across domains and datasets for the preliminary baseline results described in Table 4.

