# OpenReview forum: "MultiHal: Multilingual Dataset for Knowledge-Graph Grounded Evaluation of LLM Hallucinations"
_ICML.cc/2026/Conference — ICML 2026 regular_

### Official Review · Reviewer_JQwU · 2026-03-08

**Soundness:** 2
**Presentation:** 2
**Significance:** 2
**Originality:** 2
**Overall Recommendation:** 4
**Confidence:** 3

**Summary:**

The authors propose MultiHal, which is a multilingual, knowledge-graph-grounded benchmark for evaluating LLM hallucinations. The dataset is formed by aggregating questions from 7 existing benchmarks, mining KG paths from Wikidata, filtering them using an LLM-as-a-judge, and translating the result into 5 European languages. Their experiments show that LLMs hallucinate less when given the context from the KG paths, as compared to direct QA.

**Compliance With Llm Reviewing Policy:**

Affirmed.

**Final Justification:**

The rebuttal addressed my main concerns.

**Key Questions For Authors:**

Please find my concerns in the "Weaknesses" section. Furthermore, how do you deal with the temporal validity of the questions? Does your dataset include questions like "Who is the current US president?"

**Limitations:**

yes.

**Strengths And Weaknesses:**

Strengths:

- The topic of focus, LLM hallucinations is quite important and timely. The solution of relying on KGs for addressing hallucinations is also a rational and well-established choice.

- Experimental details are provided properly. Computation times, CO2 emissions, costs, and human audit details are all reported, which is good scientific practice.

- The scale of the dataset (more than 7000 questions with 48 domains) and the permissive licensing terms are proper for a dataset.

Weaknesses:

- The idea of using KGs to reduce LLM hallucinations and the methodology used for creating the dataset are both well-established. I struggle to find the true novelty of the work, except that the scale and breadth of the datapoints are greater than those of exising baselines.

- The language diversity is rather narrow. I appreciate the fact that the dataset is designed to be multilingual, but the five chosen languages (Spanish, French, Italian, Portuguese, German) are all high-resource languages. Low-resource languages, where hallucination problems are most severe by the authors' own admission, are entirely absent.

- I'm concerned about the quality of translations. Translations are done using an automated model (NLLB-200), and human audits reveal concerning interannotator agreement scores. For instance, for Italian (IAA binarized = 0.45) and Spanish (IAA binarized = 0.36). These are rather modest agreements for a benchmark that will be used to evaluate model factuality.

- There are some minor issues such as the poor quality of Figure 1 and two repeated words, "is" in line 182.

---

> ### Author Rebuttal · Authors · 2026-03-31
>
> Dear Reviewer JQwU
>
> Thank your for your review and please find our responses written below!
>
> # Weaknesses
> - *The idea of using KGs..*
>   1. We outline that prior to this work, researchers developing KG-RAG systems lack a standardized, multilingual testbed built upon the community's most trusted hallucination benchmarks (e.g., TruthfulQA, HaluEval). By structurally mapping these text-centric datasets to 25.9k high-quality Wikidata paths, we provide the testbed necessary for direct, apples-to-apples comparisons between vanilla QA, text-based RAG, and KG-based RAG settings. Furthermore, our method of mining, filtering, and validating the KG paths demonstrates the effectiveness of an LLM-as-a-judge for graph data which serves as a secondary contribution and a scalable blueprint for the community.
>
> - *The language diversity..*
>   2. We agree that covering low resource languages is an important next step. However, we outline that the primary goal of MultiHal is to establish a rigorous, controlled baseline for KG-grounded evaluation without introducing obfuscations in our methodology. By focusing on the five well-supported Indo-European languages via Nllb-200, we ensured that the translation quality, the sentence embedding and NLI evaluation models, and the generative LLMs all possessed reasonable baseline capabilities and that we do not risk introducing biases from specialized models for lower-resource languages. Expanding to low-resource languages would make it difficult to disentangle whether a performance drop is due to poor KG-path integration, poor translation, or an evaluated LLM’s inherent language deficit. We explicitly acknowledge this typological limitation in Section 9 and hope MultiHal serves as a foundation for future expansions.
>
> - *I'm concerned about the translations..*
>   3. The IAA scores may indeed be modest, also considering that we have a small sample size of annotators per language. However, we argue that the ultimate test of the translations' quality is their extrinsic utility in the downstream task. Despite the modest human agreement on the exact quality rating, the translated paths preserved the factual semantic signal for the LLM as demonstrated by our empirical results. Across all evaluated metrics, models (Gemini, Llama, GPT-4o Mini), and languages, the KG-RAG setting consistently (Tables 5, 11, 15 and Figure 3) and significantly (Table 13) outperforms the vanilla QA setting. If the translations were fundamentally broken or lacked signal, the models would have hallucinated more or failed to improve over the baseline. Our empirical results conclusively prove that the translated paths are highly useful for answering the benchmark questions.
>
> - *There are some minor issues*
>   4. We will fix this for camera-ready, thank you for the diligence!
>
> ---
>
> # Questions
> - *How do you deal with the temporal..*
>   1. Our dataset is indeed static and does not update in real-time. Regarding the inclusion of temporal questions, yes, the benchmark does include questions such as "Who is the current president of the United States?" (which we explicitly list as an example in Appendix I, Listing 13). By freezing the KG paths at a specific timestamp, MultiHal provides a controlled environment to evaluate whether a model can correctly ground its answer in the provided external context (the KG path). This setup is consistent with standard RAG scenarios, where models are expected to base their responses on the retrieved context even when it may be temporally misaligned with real-world facts. Thus, the task evaluates context faithfulness rather than real-time factual correctness. We acknowledge that temporal drift is an inherent limitation of static benchmarks. To address this, we release both the dataset and construction pipeline publicly, enabling re-generation or updating of the KG paths to reflect newer timestamps. This makes MultiHal extensible and adaptable for future temporal evaluations.
>
> ---
> # Final Remarks
> Thank you again for reviewing our work! If you find our answers addressing your concerns, we would highly appreciate if this would reflect back in your review ratings. Otherwise please let us know how we can follow-up.

---

> > ### Author Rebuttal · Reviewer_JQwU · 2026-04-04
> >
> > Thank you for the rebuttal. Most of my concerns have been addressed, except for my question about the translation quality and modest IAA scores. I’m not convinced that because the KG-RAG results improved, the translations must be good enough. My concern was about your benchmark's integrity. If a benchmark has modest human agreement, it is difficult to trust it as a gold standard for future researchers. How can it be ensured that models won't overfit to translation noise or artifacts present in this version of the dataset?

---

> > > ### Author Response · Authors · 2026-04-06
> > >
> > > Dear Reviewer JQwU,
> > >
> > > Thank you for engaging with us with the follow-up! We agree that downstream performance does not *necessarily* equate
> > > to linguistic perfection although we would outline that it certainly is influenced and our empirical
> > > results **at minimum** support a *good-enough* linguistic level. In either case we would break down
> > > our response as follows.
> > >
> > > ---
> > >
> > > 1. **Why the human IAA for evaluating translation accuracy is modest (The Nature of KG Triples).**
> > >
> > > We acknowledge that the IAA
> > > scores are modest for two of the translated languages (Spanish and Italian) and significant for German and
> > > Portuguese, which means that our IAAs have a variance when determining the translation quality.
> > >
> > > As documented in MT literature [1-2], human raters frequently disagree on how to grade non-fluent or syntactically awkward text, the IAA variance is a common problem for the field and the human-annotation methodologies. Especially paper [2] outlines that human disagreements
> > > are more emphasized when dealing with non-standard linguistic phenomena. In our case, we
> > > are presenting raters with disjointed triples which is not fluent text (e.g., France ; capital ;
> > > Paris).
> > >
> > > Also, we did not have reference translations that we could present to our annotators since the translation work hadn’t been done, therefore we had to rely on our annotators bilingualism. While all the annotators, at minimum, speak professional-level English and their corresponding translation language, the lack of reference translation definitely did not help them make consistent judgments. We have not done an in-depth follow up study on this but this is how we can explain the IAAs. Please let us know if this addresses your concerns or if we can follow-up.
> > >
> > > 2. **Mitigating the Risk of Overfitting.**
> > >
> > > MultiHal is explicitly designed as a zero-shot benchmark. Because the evaluated models (Gemini, Llama, GPT) are not being trained on this data, they would not be parametrically overfitting to the translation artifacts. We also outline that our empirical results showcase that all the tested model families generalize well on all the languages.
> > >
> > > ---
> > >
> > > **Final remarks:** Thank you for your diligent comments, please find the two references below and we hope this addresses your concerns!
> > >
> > > [1] Callison-Burch, Chris, et al. "(Meta-) evaluation of machine translation." Proceedings of the Second Workshop on Statistical Machine Translation. 2007.
> > >
> > > [2] Popović, Maja. "Agree to disagree: Analysis of inter-annotator disagreements in human evaluation of machine translation output." Proceedings of the 25th Conference on Computational Natural Language Learning. 2021.

---

### Official Review · Reviewer_qWzq · 2026-03-11

**Soundness:** 2
**Presentation:** 3
**Significance:** 3
**Originality:** 3
**Overall Recommendation:** 5
**Confidence:** 1

**Summary:**

MultiHal addresses a gap in the hallucination evaluation landscape: existing benchmarks are English-only and rely on unstructured text for grounding, leaving Knowledge Graph integration and multilingual evaluation largely unexplored. The authors aggregate seven established hallucination benchmarks, mine ~140k KG paths from Wikidata using entity linking via Falcon 2.0, filter them to 25.9k high-quality paths through a two-step LLM-as-a-judge pipeline, and translate the resulting dataset into five European languages. Baseline experiments across three LLMs consistently show that providing mined KG paths as in-context knowledge (KG-RAG) outperforms vanilla QA across all languages and all three evaluation metrics, semantic similarity, NLI entailment, and hallucination detection.

**Compliance With Llm Reviewing Policy:**

Affirmed.

**Final Justification:**

The analysis of LLM judge noise across 31 domains (Table R1) clarifies that false positives aren't clustering in specific areas, which validates the per-domain comparisons in Table 16. I recommend including this in the appendix. Regarding multilingual evaluation, I find the use of NLI as a proxy for hallucination detection reasonable given current tool limitations, and the consistent gains across both metrics support the claims. I'm satisfied with these clarifications.

**Key Questions For Authors:**

1. On the LLM judge as benchmark gatekeeper. The ~11% false positive rate means a non-trivial share of retained paths are low quality by human standards. Is this noise uniformly distributed across domains, or does it cluster in specific subsets, particularly TruthfulQA and HaluBench, where Wikidata coverage is already acknowledged to be weak? If noise clusters there, per-domain comparisons in Table 16 could be misleading. Showing the noise is roughly uniform, or controlling for it, would directly address the main soundness concern in this review.

2. On multilingual evaluation completeness. Given that HHEM-2.1 is English-only, what evidence supports the claim that KG-RAG reduces hallucinations specifically in the non-English subsplits? NLI entailment is a reasonable proxy but is not the same as hallucination detection. Is there a multilingual hallucination detection model that could be applied to at least a subset of the non-English data to validate this?

3. Quick comment. There’s a typo on page 4: “The answer type is is denoted”. Please, correct it.

**Limitations:**

Partially addressed. Section 9 is upfront about typological diversity, multi-prompt evaluation, and the simplicity of the KG-RAG baseline. What is missing is any discussion of using a noisy LLM judge to define benchmark quality rather than to evaluate outputs, which is the most consequential design choice in the paper.

**Strengths And Weaknesses:**

Strengths

- The combination of KG grounding and multilinguality in a single hallucination benchmark is a distinct contribution. The closest comparators (MintakaQA, MKQA) offer either multilingual coverage or Wikidata annotations, but not full KG paths.

-The KG-RAG improvements are statistically significant across all 18 language-model combinations and consistent across the large-scale subsets (HaluEval, DefAn, SimpleQA, Shroom2024) rather than driven by a single domain, which gives confidence that the mined paths carry real signal.

- Disclosure standards are high: CO₂ estimates, per-stage compute costs, false positive/negative rates for the judge, and human translation audits with IAA scores are all reported.

Weaknesses
- The LLM judge is used not merely to evaluate model outputs but to define the benchmark itself. An 11% false positive rate (Table 4) means roughly 1 in 9 retained paths is noise by human standards. The authors compare this favourably to QA dataset noise (20–30%), but those datasets are not used to gate what counts as a valid evaluation signal, the comparison does not fully hold.

- All five target languages are Indo-European and high-resource (German, French, Italian, Portuguese, Spanish). This means the benchmark says little about whether KG-RAG helps for languages that are structurally very different from English, or for languages where Wikidata coverage is sparser, which are arguably the cases where hallucination mitigation matters most.

---

> ### Author Rebuttal · Authors · 2026-03-31
>
> Dear Reviewer qWzq,
>
> Thank you for your review! Please find our responses below.
>
> # Weaknesses
> - *The LLM judge is used..*
>   1. We acknowledge the 11\% noise estimate in terms of false positives with respect to the preliminary baseline subsplit. However, we argue that the overwhelming density of the signal completely outweighs this noise. We demonstrate this empirically in our baseline experiments: when generative LLMs are conditioned on these filtered paths (KG-RAG condition), we observe consistent improvements across all evaluated metrics, models, and languages (Figure 3, Tables 5 and 6). Furthermore, these improvements are statistically significant as demonstrated by our Cramer-von Mises tests (Table 13). If the 11\% noise were actively detrimental to the evaluation signal, we would not see strictly positive deltas. Therefore, while not perfectly noise-free, the pipeline effectively isolates a strong and valid evaluation signal.
>
> - *All five target languages are Indo-European..*
>   2. As noted in our Answer #2 to Reviewer JQwU under *weaknesses*, establishing a rigorous, controlled baseline was our primary objective through validating on languages where the translation model (Nllb-200), evaluation metrics and generative LLMs have reasonable multilingual capabilities.
>
> ---
>
> # Key Questions
> - *On the LLM judge as benchmark..*
>   1. We had a look at the data and after analyzing the false positives for the preliminary baseline subsplit, we found them to be distributed across 31 out of 58 domains. Crucially, this distribution is approximately uniform and does not disproportionately cluster in the TruthfulQA or HaluBench subsets. While we explicitly acknowledge in Section 6 that domains like TruthfulQA and HaluBench pose distinct challenges due to temporal, suggestive, or reasoning-heavy questions, the rating assignment to the corresponding paths themselves are sound. You can inspect the distribution in the Table below and we are happy to further expand upon this and include as an additional appendix for the camera-ready version.
>
> | Domain | Weighted Disagreements | Number of Disagreements | Total Examples | Source Dataset |
> | - | - | - | - | - |
> | conspiracies | 0.21 | 3 | 14 | tqa_gen |
> | indexical error: identity | 0.0909 | 1 | 11 | tqa_gen |
> | indexical error: time | 0.0909 | 1 | 11 | tqa_gen |
> | indexical error: location | 1.0 | 1 | 1 | tqa_gen |
> | distraction | 0.1538 | 2 | 13 | tqa_gen |
> | advertising | 0.0833 | 1 | 12 | tqa_gen |
> | religion | 0.0909 | 1 | 11 | tqa_gen |
> | history | 0.1148 | 7 | 61 | tqa_gen |
> | confusion: people | 0.1 | 1 | 10 | tqa_gen |
> | confusion: places | 0.5 | 6 | 12 | tqa_gen |
> | misinformation | 0.25 | 1 | 4 | tqa_gen |
> | entertainment | 0.3611 | 26 | 72 | defan |
> | nobleprize | 0.2222 | 16 | 72 | defan |
> | sports | 0.2314 | 28 | 121 | defan |
> | worldorg | 0.3947 | 30 | 76 | defan |
> | qsranking | 0.0921 | 7 | 76 | defan |
> | wk | 0.1466 | 28 | 191 | felm |
> | general | 0.0088 | 1 | 114 | halubench |
> | pubmed | 0.0706 | 6 | 85 | halubench |
> | finance | 0.0083 | 1 | 121 | halubench |
> | covid | 0.0938 | 3 | 32 | halubench |
> | qa | 0.1503 | 72 | 479 | halueval |
> | N/A | 0.0217 | 10 | 461 | shroom2024 |
> | science and technology | 0.0588 | 3 | 51 | simpleqa |
> | geography | 0.1458 | 7 | 48 | simpleqa |
> | art | 0.25 | 13 | 52 | simpleqa |
> | politics | 0.0678 | 4 | 59 | simpleqa |
> | other | 0.0784 | 4 | 51 | simpleqa |
> | tv shows | 0.1154 | 6 | 52 | simpleqa |
> | music | 0.125 | 6 | 48 | simpleqa |
> | video games | 0.2 | 9 | 45 | simpleqa |
>
> - *On multilingual evaluation..*
>   2. Correct that NLI entailment, while a strong proxy, is not strictly identical to hallucination detection. At the time of executing this work, we were not aware of established, production-grade open-source hallucination detection models with reliable multilingual capabilities on par with HHEM-2.1 (which is restricted to English). While we could employ a high-tier closed-source LLM as a multilingual hallucination judge, properly validating that specific model's cross-lingual hallucination detection accuracy and bias across five distinct target languages would constitute a massive, separate research study in itself. Because of this limitation, we supplemented semantic similarity with the addition of NLI evaluation. Given the consistent improvements across both of these rigorous proxy metrics for the non-English splits, we maintain high confidence in the factuality improvements, though we readily agree that dedicated multilingual hallucination detection models are an important direction of NLP research.
>
> - *Quick comment..*
>   3. We will correct this for the camera-ready! Thank you!
>
>
> ---
> # Final Remaks
> Thank you again for the review! If you find our responses addressing your concerns, we would appreciate if you could reflect this in your review scores! Otherwise please let us know how we can follow-up.

---

> > ### Author Rebuttal · Reviewer_qWzq · 2026-04-07
> >
> > My two key questions have been addressed satisfactorily:
> >
> > - LLM judge noise distribution: The authors analyzed the false positives across the preliminary baseline subsplit and found them distributed across 31 out of 58 domains, with no disproportionate clustering in TruthfulQA or HaluBench. This directly addresses my concern that per-domain comparisons in Table 16 could be misleading. I appreciate the offer to include this analysis as an appendix in the camera-ready and would encourage the authors to do so.
> >
> > - Multilingual hallucination detection: The authors honestly acknowledge that NLI entailment is a proxy rather than a direct substitute for hallucination detection. Given the current lack of established multilingual hallucination detection models comparable to HHEM-2.1, supplementing semantic similarity with NLI evaluation is a reasonable approach. The consistent improvements across both proxy metrics for non-English splits support the paper's claims.

---

> > > ### Author Response · Authors · 2026-04-08
> > >
> > > Dear Reviewer qWzq,
> > >
> > > We highly appreciate you engaging with out rebuttal and we're happy to hear that our rebuttal has addressed your key questions!
> > >
> > > We thank you for your time and effort into reviewing our work!

---

### Official Review · Reviewer_MeNN · 2026-03-12

**Soundness:** 3
**Presentation:** 3
**Significance:** 3
**Originality:** 3
**Overall Recommendation:** 4
**Confidence:** 4

**Summary:**

The paper introduces MultiHal, a novel multilingual benchmark designed to evaluate and mitigate hallucinations in Large Language Models (LLMs) through Knowledge Graph (KG) grounding. While existing benchmarks are primarily English-centric and rely on unstructured text, MultiHal bridges these gaps by mining 140k KG paths (pruned to 25.9k high-quality paths) from Wikidata to provide structured factual context. The dataset covers five European languages (German, Italian, French, Portuguese, and Spanish) in addition to English. The authors propose a unified framework for entity linking, KG path mining, and quality control using LLM-as-a-judge. Baseline experiments using models like Gemini 2.0 and Llama 3.3 demonstrate that incorporating KG paths in a Retrieval-Augmented Generation (RAG) setting significantly improves semantic similarity and reduces hallucinations across all tested languages.

**Compliance With Llm Reviewing Policy:**

Affirmed.

**Key Questions For Authors:**

- The authors mention that suggestive questions in TruthfulQA require logical reasoning over KG paths. How well do current models perform on these specifically compared to simple fact-retrieval questions?
- Given the reliance on Nllb-200 and Wikidata, how feasible is it to extend MultiHal to languages with significantly less KG coverage?
- Authors used 8 native speakers for audits. Was there a significant disagreement between human judges and the GPT-4o Mini judge in specific domains (e.g., finance or health)?
- Since RAG is sensitive to prompt formatting, did the authors experiment with different prompt templates for injecting the KG paths, or only the "concise and explicit" format described?

**Limitations:**

- The benchmark is focused on single-round question answering and does not address multi-round dialogue or text summarization.
- Performance is lower in specialized domains like medicine (Pubmed/Covid) because the pipeline relies on the general-domain Wikidata rather than domain-specific KGs.
- The current pipeline lacks explicit reasoning integration (e.g., Think-on-Graph), which limits its effectiveness for complex reasoning-based queries.
- The study does not include multi-prompt evaluation to account for model sensitivity to minor prompt variations.

**Strengths And Weaknesses:**

Strengths:

- Addresses a significant gap in current hallucination benchmarks, which are predominantly English-only, by providing high-quality translations for five additional languages.
- Leverages Knowledge Graphs rather than just unstructured text, which provides minimal linguistic overhead and enables better explainability and factual tracing.
- Employs a multi-step pipeline including deduplication, redundant entity matching (Falcon 2.0 + DBpedia + Wikipedia), and a two-step LLM-as-a-judge scoring system (validated by human audits) to ensure path quality.
- Aggregates and repurposes data from seven established benchmarks (e.g., TruthfulQA, HaluEval, SimpleQA), resulting in a broad spectrum of question types and 48 distinct domains.

Weaknesses:

- The multilingual support is restricted to Western European languages, which the authors acknowledge as a limitation. Low-resource or non-Latin script languages are not included.
- The authors note significant noise when using Falcon 2.0 for entity linking, leading to many low-quality paths that require heavy filtering.
- The quality control relies heavily on GPT-4o Mini, which may introduce its own biases or hallucinate during the selection/scoring process, though the authors attempt to mitigate this with multiple passes and overlapping selections.
- Sentence embedding metrics used for evaluation (Semantic Similarity) sometimes penalize concise, correct model responses if the ground-truth contains question repetitions.
- The benchmark relies on a specific Wikidata cut-off (April 2025), which may not account for the evolving nature of facts in temporal questions.

---

> ### Author Rebuttal · Authors · 2026-03-31
>
> Dear Reviewer MeNN,
>
> Thank you for the diligent review! Please see our responses below
>
> # Weaknesses
> - *The multilingual support is restricted...*
>   1. Thank you for this point, please see our Answer #2 to Reviewer *JQwU* under *Weaknesses*
>
> - *The authors note significant...*
>   2. It is true that raw entity linking from text inherently introduces noise, which is exactly why our core contribution includes a rigorous, multi-step quality assurance pipeline. Our pipeline includes steps such as removing circular subject-object pairs, processing Top-3 entity candidates, querying and mapping from DBPedia to Wikidata, two pass LLM-judge with additional steps for ensuring robustness. All the aforementioned filtering steps we see as a strength rather than a weakness. We believe this to be an effective recipe for the community. This pipeline successfully distilled 140k noisy candidate paths down to a high-quality subset of 25.9k paths. The effectiveness of this filtering is empirically proven by the consistent, statistically significant improvements across all evaluated metrics, models, and languages when these filtered paths are injected (Figure 3, Tables 5 and 6).
>
> - *The quality control relies heavily..*
>   3.  We took a highly conservative approach to mitigate LLM-judge hallucinations and biases. During the selection phase additional to multiple passes with randomized input ordering and overlapping selections between passes, if a selected path generated by GPT-4o Mini that did not have a perfect exact-match from the original candidate pool, it was discarded to prevent hallucinations. Regarding ratings, we computed IAA scores between human judge and GPT-4o Mini, achieving a Cohen-Kappa score of 0.62 which is a substantial agreement. Furthermore, Table 4 reports an 11\% False Positive rate, demonstrating that while some noise exists, it is well within acceptable margins for QA datasets. Finally, if the paths were severely biased toward GPT-4o Mini, we would not have observed the statistically significant semantic score improvements (Table 13) when testing the paths on different model families, such as Llama 3.3 70B and Gemini 2.0 Flash.
>
> - *Sentence embedding metrics*
>   4. You are correct about the inherent limitations of sentence embeddings, which we also explicitly discuss in Section 6. To mitigate this exact issue, we did not rely on semantic similarity alone; we triangulated our evaluation using NLI (mDeBERTa) and hallucination detection (HHEM-2.1). Across all three metrics, the KG-RAG improvements hold. Regarding the Wikidata cut-off and temporal questions (e.g., "Who is the current president"), we also highlight this challenge in Section 6.
>
> - *The benchmark relies...*
>   5. Please see our answer to Reviewer JQwU under *Key Questions*
>
> ---
>
> # Key Questions
> - *The authors mention that..*
>   1. In our methodology, logical reasoning over a KG path is inherently captured by paths rated with a Quality Score of 4. As defined in our LLM-judge instructions (Appendix E, Listing 9). Because these paths contain implicit rather than explicit answers, the evaluated LLMs are required to perform a degree of logical reasoning over the structured data to arrive at the correct answer.
>
> - *Performance is lower in...*
>   2. We believe it is highly feasible because of our pipeline's design. Our methodology performs the KG querying, path extraction, and path filtering in English. We only utilize Nllb-200 at the very end to translate the finalized question-answer pairs and their corresponding English KG paths into the target languages. Therefore, extending MultiHal to new languages depends entirely on the capability of the translation model to accurately translate the English triples and text. Additionally we ouline that Wikidata also provides curated multilingual labels, though they may be sparsely available.
>
> - *The current pipeline..*
>   3. We would like to clarify a slight misunderstanding regarding the human audits. The 8 native speakers (2 per language) were utilized exclusively to audit the linguistic quality of the Nllb-200 translations (Appendix P, Table 19), not to evaluate the GPT-4o Mini path judge. The evaluation of the GPT-4o Mini judge against human judgment (which yielded the 0.62 Cohen-Kappa agreement) was conducted on the English preliminary baseline subsplit (Table 3). Please see our response to Reviewer qWzq regarding noise distribution.
>
> - *Since RAG is..*
>   4. For this baseline evaluation, we restricted our experiments to the single, straightforward prompt templates detailed in Appendix F. Our primary objective was to validate the quality and utility of the mined paths themselves, rather than to find the upper-bound performance achievable through prompt engineering.
>
> ---
>
> # Final Remarks
> Thank you for the diligent review, if our responses have addressed your concerns, we would appreciate if you would increase your review score. Otherwise please let us know how we can further follow-up

---

### Decision · Program_Chairs · 2026-04-30

**Decision:**

Accept (regular)

**Comment:**

Summary:

This paper proposes MultiHal, a KG-based, multilingual, multi-hope benchmark for generative text evaluation. The authors mined 140k KG-paths from open-domain KGs and pruned noisy ones, retaining a high-quality subset of 25.9k. The experiments show that using MultiHal as the evaluation baseline led to ~0.12-0.36 improvements in semantic similarity scores, ~0.16-0.36 improvements in NL entailment, and ~0.29-0.42 improvements in hallucination detection in KG-RAG, relative to vanilla QA, across multiple languages and models.

Justifications:

MultiHal addresses the gap in the current hallucination benchmarks, which mainly focus on English. The authors provided a detailed and comprehensive rebuttal to all reviewers. Reviewer qWzq acknowledged that the rebuttal fully addressed his questions and concerns and recommended acceptance of the paper. Two other reviewers recommended weak acceptance. Reviewer MeNN did not engage in the rebuttal, so I assume the authors' answers are sufficient. Reviewer JQwU acknowledged that the authors' response partially addressed his questions and concerns, but did not acknowledge or respond to the authors' follow-up, so I assume the answers are sufficient. I recommend the authors incorporate all reviewers's comments and feedback during the rebuttal into the final version of the paper.